# Network-wide abnormalities explain memory variability in hippocampal amnesia

**Georgios PD Argyropoulos[1]\*, Clare Loane[1,2], Adriana Roca-Fernandez[1], Carmen Lage-Martinez[1,3], Oana Gurau[1], Sarosh R Irani[4], Christopher R Butler[1]**

[1]Memory Research Group, Nuffield Department of Clinical Neurosciences, University of Oxford, Oxford, United Kingdom; [2]Institute of Cognitive Neuroscience, University College London, London, United Kingdom; [3]Valdecilla Biomedical Research Institute, University Hospital Marqués de Valdecilla, Santander, Spain; [4]Oxford Autoimmune Neurology Group, Nuffield Department of Clinical Neurosciences, University of Oxford, Oxford, United Kingdom

**Abstract** Patients with hippocampal amnesia play a central role in memory neuroscience but the neural underpinnings of amnesia are hotly debated. We hypothesized that focal hippocampal damage is associated with changes across the extended hippocampal system and that these, rather than hippocampal atrophy per se, would explain variability in memory between patients. We assessed this hypothesis in a uniquely large cohort of patients (n = 38) after autoimmune limbic encephalitis, a syndrome associated with focal structural hippocampal pathology. These patients showed impaired recall, recognition and maintenance of new information, and remote autobiographical amnesia. Besides hippocampal atrophy, we observed correlatively reduced thalamic and entorhinal cortical volume, resting-state inter-hippocampal connectivity and activity in posteromedial cortex. Associations of hippocampal volume with recall, recognition, and remote memory were fully mediated by wider network abnormalities, and were only direct in forgetting. Network abnormalities may explain the variability across studies of amnesia and speak to debates in memory neuroscience.
DOI: https://doi.org/10.7554/eLife.46156.001

**\*For correspondence:**
georgios.argyropoulos@ndcn.ox.ac.uk

## Introduction

Ever since the first report of the famous patient H.M. (*Scoville and Milner, 1957*), hippocampal (HPC) amnesia has played a fundamental role in the neuroscience of human memory (*MacPherson and Della Sala, 2019*). Studies of amnesic patients have, however, often produced inconsistent results. Such studies have been criticized for attributing behavioral deficits solely to the HPC (*Squire and Wixted, 2011*), despite the fact that focal damage may trigger more subtle structural and functional abnormalities across an extended HPC system, involving the thalamus and cingulate cortex (*Aggleton, 2014*). Indeed, broader network abnormalities are well documented in other 'focal' conditions [e.g. ischemic stroke (*Carter et al., 2010*; *Veldsman et al., 2018*)] and have been demonstrated in small case series of amnesic patients (*Hayes et al., 2012*; *Henson et al., 2016*; *Rudebeck et al., 2013*). These abnormalities may be important in understanding memory impairment in neurological disease (*Addis et al., 2007*) but their explanatory potential is under-explored.

We propose that network abnormalities are key to understanding inconsistencies that underpin central debates about HPC function in amnesia research. One such debate centers on the dissociation of neurocognitive processes underlying fundamental aspects of anterograde retrieval. Some

authors hold that recall relies on HPC-dependent recollection processes, whereas recognition additionally draws on familiarity, which is mediated by non-HPC structures within the medial temporal lobe (MTL) (*Brown and Aggleton, 2001*; *Kafkas et al., 2016*). Others argue that recollection and familiarity do not dissociate within the MTL (*Wixted and Squire, 2010*). Related to the above, some theorists posit that HPC processing is material-specific, whereby the HPC holds a privileged role in processing scenes or spatial stimuli and this function underpins its involvement in episodic memory and navigation (*Maguire and Mullally, 2013*; *Graham et al., 2010*). An opposing view holds that the primary role of the HPC is the material-independent maintenance of information over a limited time period (*Kim et al., 2015*). A second, long-standing debate (*Cassel and Kopelman, 2019*), pertains to the presence of abnormal (accelerated) forgetting in HPC amnesia. Whereas some reports have provided evidence for abnormal forgetting after MTL damage (*Huppert and Piercy, 1979*; *Isaac and Mayes, 1999*), other studies have reported normal forgetting rates (*Freed et al., 1987*; *Kopelman, 1985*; *McKee and Squire, 1992*). A third debate pertains to the HPC role in remote memory. The 'standard consolidation theory' holds that newly-acquired memories initially depend upon the HPC but that, through a process of 'consolidation', memory traces gradually become HPC-independent and are stored in and retrieved by neocortical structures alone (*Hardt and Nadel, 2018*; *Squire et al., 2015*). In contrast, 'multiple trace theory' proposes that the HPC is necessary for vivid episodic recollection irrespective of the age of the memory (*Moscovitch et al., 2005*).

A central problem for addressing these debates with neuropsychology and, more specifically, for determining the relevance of wider brain network abnormalities, is that HPC amnesia is very rare. Consequently, most studies rely upon just a handful of amnesic cases and are thus susceptible to confounds such as impairment severity, measurement noise and individual differences (*Lambon Ralph et al., 2011*; *Lambon Ralph et al., 2002*). Larger cohort studies would partially overcome these confounds and capitalize on case heterogeneity to map behavior to brain structure and function.

In order to explore the impact of HPC damage on wider brain networks and determine the impact of these abnormalities upon memory, we studied a uniquely large cohort (n = 38) of patients after autoimmune limbic encephalitis (LE). In its acute phase, this highly consistent clinical syndrome classically causes focal HPC damage, seen on MRI as high T2 signal (*Finke et al., 2017*; *Kotsenas et al., 2014*; *Loane et al., 2019*; *Malter et al., 2014*). Although most patients respond well to early immunosuppressive therapy (*Finke et al., 2017*; *Irani et al., 2011*; *Thompson et al., 2018*), some subsequently develop HPC atrophy and persistent cognitive impairment, the most prominent aspect of which is anterograde amnesia (*Butler et al., 2014*; *Finke et al., 2017*; *Irani et al., 2013*; *Loane et al., 2019*; *Malter et al., 2014*). Autoimmune LE patients are often included in studies of HPC amnesia (*Hassabis et al., 2007*; *Henson et al., 2017*; *Henson et al., 2016*; *Maguire et al., 2006*). Post-mortem studies demonstrate focal HPC damage (*Khan et al., 2009*; *Park et al., 2007*), contrasting with the wider damage typically found following other encephalitides [e.g. herpes simplex encephalitis (*Damasio and Van Hoesen, 1985*; *Gitelman et al., 2001*)] or global ischemia/anoxia (*Huang and Castillo, 2008*), conditions that have also been used as models of human HPC function (for discussion, see *Squire and Zola, 1996*). Animal models have also shown focal limbic involvement (HPC, amygdala) (*Tröscher et al., 2017*). Patients' varying degrees of residual symptom severity offered us the opportunity to examine the relationships between cognition and brain structure and function.

Patients (n = 38) and healthy controls (n = 41) were assessed with a thorough battery of neuropsychological tests of episodic memory (for verbal and visual material) and other cognitive functions, in order to identify the tasks in which patients showed impaired performance. The tests we employed are robust, widely used, well-normed, and largely model-free (see below). Using manual volumetry of MTL regions, as well as whole-brain, voxel-based investigations of structural MRI, we identified regions of gray matter (GM) volume reduction in patients as compared with healthy controls. Using resting-state fMRI (rsfMRI), we also identified resting-state functional abnormalities in patients with respect to both segregation and integration, that is in terms of hemodynamic activity in local regions (resting-state amplitude of low-frequency fluctuations; rsALFF) and functional connectivity between regions (resting-state functional connectivity; rsFC). Finally, we investigated the relationships between brain abnormalities and cognitive deficits. We predicted that: i) HPC damage (documented to occur focally in autoimmune LE) would be accompanied by remote abnormalities in hubs of the extended HPC system, including the HPC-diencephalic-cingulate network (*Aggleton, 2014*;

*Bubb et al., 2017*); ii) variation in the extent of these network-wide abnormalities across patients would explain variability in memory impairment over and above HPC atrophy; iii) Moreover, on our assumption that HPC damage leads to broader network disruption, we predicted that relationships between HPC atrophy and memory impairment would be mediated by this network disruption. This prediction is in line with recent functional imaging and lesion studies that emphasize the importance of broader networks in episodic memory (*Cook et al., 2015*; *Henson et al., 2016*; *Inhoff and Ranganath, 2017*; *Schedlbauer et al., 2014*; *Watrous et al., 2013*); iv) taking these abnormalities into account would enable us to identify which particular aspects of memory impairment are a direct function of HPC atrophy.

## Results

Our analysis approach is summarized in *Figure 1*. We first (1) identified cognitive deficits by comparing the performance of patients and healthy controls on a broad range of neuropsychological tests. We then (2) identified regions in which patients showed structural and (3) functional abnormalities relative to healthy controls. We then (4) examined the relationship between structural/functional brain abnormalities and memory impairment across patients.

### Neuropsychological assessment

The patients included in this study (n = 38) had been diagnosed with autoimmune LE, and were recruited post-acutely, after reassessment by an experienced neurologist (CRB) prior to study inclusion (see Materials and methods).

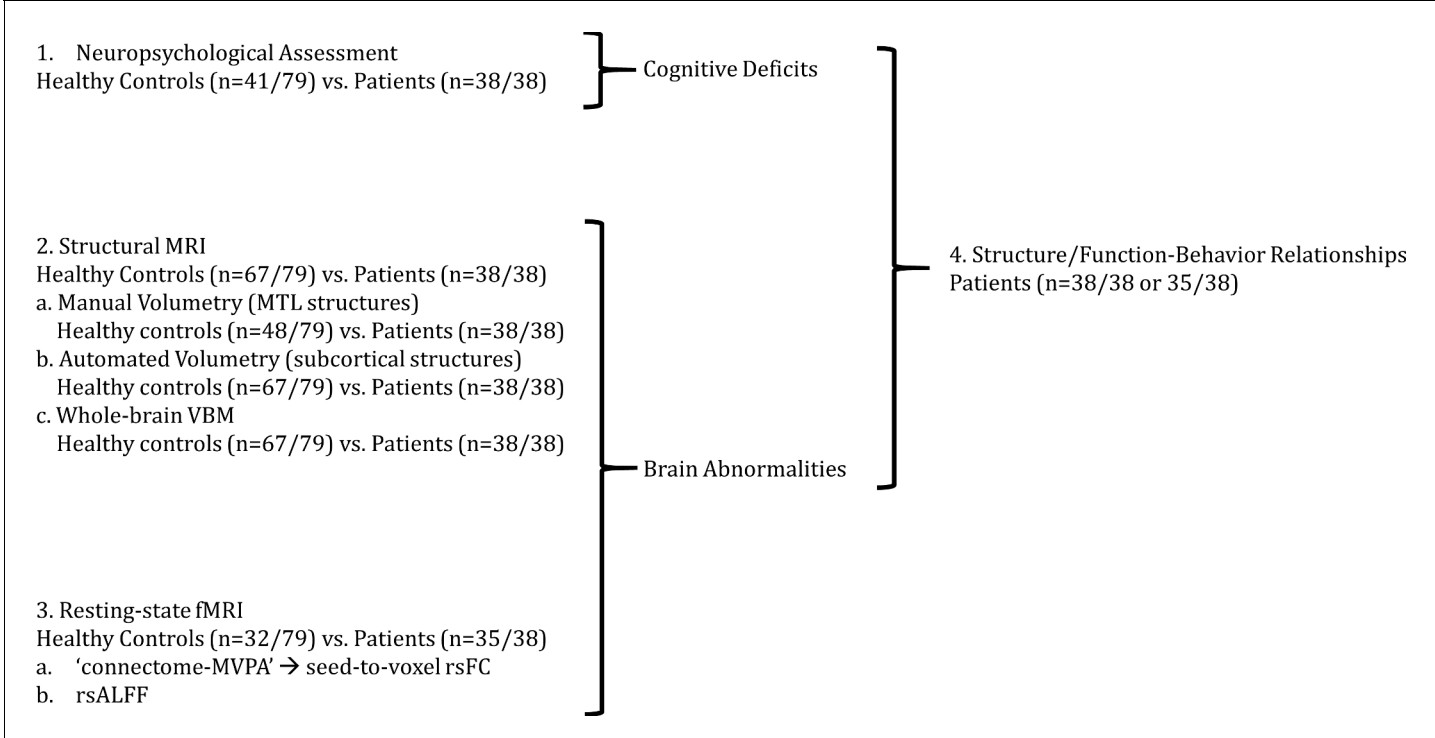

**Figure 1.** Outline of Results Section. We first (1) identified cognitive deficits by comparing patients with healthy controls in a broad range of tests of neuropsychological assessment. We identified regions in which patients showed (2) reduced gray matter volumes and (3) resting-state functional connectivity and activity relative to healthy controls; (4) we also identified relationships between structural/functional abnormalities and performance in tests in which patients showed impairment as compared with healthy controls; 'connectome-MVPA': connectome 'multi-variate pattern analysis' (*Whitfield-Gabrieli and Nieto-Castanon, 2012*); MRI: Magnetic Resonance Imaging; MTL: medial temporal lobe; n: number of participants; rsALFF: resting-state amplitude of low-frequency fluctuations; rsFC: resting-state functional connectivity; VBM: voxel-based morphometry.
DOI: https://doi.org/10.7554/eLife.46156.002

Patients did not differ from healthy controls (n = 41; matched for age and M:F ratio) in premorbid intelligence, semantic memory and language, visuomotor and executive function. Their cognitive profile was characterized by highly focal episodic memory impairment. Patients were impaired in both recall and recognition of both verbal and visual material (including scenes), and showed pronounced forgetting of visual material – verbal forgetting was marginally impaired. A clear exception was patients' preserved face recognition memory. They also showed impaired remote autobiographical memory for both childhood and early adulthood epochs, but preserved remote personal semantic memory. Although they showed higher scores on the depression sub-scale of the Hospital Anxiety and Depression Scale (HADS) (*Zigmond and Snaith, 1983*), the median score was well below the clinical cut-off, and none of the patients scored within the severe range (mild range: three patients; moderate range: four patients; non-case range: rest of patients and all controls) (*Table 1*).

For further correlational analyses, and in order to minimize the contribution of measurement error and maximize the generalizability of our findings, we derived three composite memory scores per participant, in relation to the aforementioned debates in the literature of HPC amnesia. These composite measures comprised scores on individual tests in which patients showed impaired performance at group level compared with controls: i) a composite score for anterograde retrieval, comprising scores on tests of verbal recall, verbal recognition, visual recall, and visual recognition, derived by averaging the corresponding standardized age-scaled scores; ii) a score for anterograde retention ('forgetting'), which was the visual forgetting score in the Doors and People test (*Baddeley et al., 1994*); iii) a remote autobiographical memory score, calculated by summing the scores on autobiographical memories for childhood and early adulthood epochs from the Autobiographical Memory Interview (*Kopelman et al., 1989*) (Supplementary Table 1 in *Supplementary file 1*).

## Structural abnormalities

### Volumetry

Consistent with the neuroradiological reports on patients' acute clinical T2-weighted MRI scans (*Table 2*) and the fact that acute HPC T2 hyperintensity and oedema are followed by post-acute HPC atrophy in autoimmune LE (*Finke et al., 2017*; *Irani et al., 2013*; *Loane et al., 2019*), volumetric analysis of patients' post-acute MRIs revealed pronounced bilateral HPC atrophy (left HPC: F = 46.02, p-corr <0.0005; right HPC: F = 63.38, p-corr <0.0005). We also observed right entorhinal (F = 10.76, p-corr = 0.0308) and left thalamic volume reduction (F = 15.41, p-corr = 0.0003; *Table 3*).

### Whole-brain VBM: GM volume

Strongly consistent with the volumetric findings above, a whole-brain VBM contrast disclosed reduced GM volume in patients' left and right HPC, as well as in the anterior/mediodorsal, and right dorsolateral regions of the thalamus (*Figure 2a*; *Table 4*).

## Functional abnormalities

We also investigated resting-state functional abnormalities in patients with respect to hemodynamic activity in local regions (resting-state amplitude of low-frequency fluctuations; rsALFF) and functional connectivity between regions, in the form of resting-state functional connectivity (rsFC) (see Materials and methods section), across the whole brain and in a data-driven fashion.

### Resting-state hemodynamic activity

In order to identify specific brain regions that show abnormal hemodynamic activity at rest in patients, similar to resting-state CBF and glucose metabolic rate in PET studies, we analyzed rsALFF, that is the intensity of slow spontaneous fluctuations of hemodynamic activity at rest across the whole brain (*Zang et al., 2007*).

As compared with healthy controls, patients showed reduced rsALFF in the posterior cingulate cortex (PCC) and the precuneus (*Figure 2b–c*). Consistent with the reciprocal connectivity of thalamic nuclei with both the HPC and the cingulate cortex (*Aggleton, 2014*; *Aggleton et al., 2010*; *Bubb et al., 2017*) and our hypothesis that HPC atrophy is followed by structural and functional abnormalities in interconnected areas within the HPC-diencephalic-cingulate networks, we observed

**Table 1.** Neuropsychological profile of autoimmune LE patients (post-acute phase) and healthy controls

| Domain | Test | Subtest | Controls M | IQR | Patients M | IQR | Test | Statistic | p-corr | cut-off score | Patients (n) | Controls (n) |
|---|---|---|---|---|---|---|---|---|---|---|---|---|
| Episodic Memory | Immediate Verbal Recall | WMS-III Logical Memory I (z) | 0.33 | 1.92 | −1.00 | 1.34 | t | 6.78 | <0.0005 | ≤ - 1.67 | 14 | 0 |
| | | Word List I (z) | 0.67 | 1.83 | −1.00 | 1.51 | U | 178.00 | <0.0005 | ≤ - 1.67 | 11 | 0 |
| | | D and P People (z) | −0.33 | 1.59 | −1.33 | 1.00 | U | 266.00 | <0.0005 | ≤ - 1.67 | 15 | 1 |
| | Delayed Verbal Recall | WMS-III Logical Memory II (z) | 0.67 | 1.67 | −2.00 | 2.00 | U | 106.50 | <0.0005 | ≤ - 1.67 | 23 | 0 |
| | | Word List II (z) | 1.33 | 1.00 | −0.67 | 1.83 | U | 179.00 | <0.0005 | ≤ - 1.67 | 3 | 0 |
| | Verbal Forgetting | D and P Verbal Forgetting (z) | 0.67 | 1.00 | −0.33 | 1.75 | U | 415.50 | 0.0584 | ≤ - 1.67 | 8 | 1 |
| | Verbal Recognition | Names (z) | 0.33 | 2.00 | −1.00 | 2.00 | t | 5.16 | <0.0005 | ≤ - 1.67 | 13 | 1 |
| | | RMT Words (z) | 1.00 | 1.51 | 0.00 | 2.26 | U | 317.50 | <0.0005 | ≤ - 1.67 | 9 | 4 |
| | | WMS-III Word List II Recognition (z) | 0.67 | 1.00 | −0.67 | 2.16 | U | 266.50 | <0.0005 | ≤ - 1.67 | 9 | 0 |
| | Immediate Visual Recall | D and P Shapes (z) | 0.67 | 1.00 | −0.84 | 2.08 | Wt | 5.78 | <0.0005 | ≤ - 1.67 | 13 | 0 |
| | | ROCFT Immediate Recall (z) | 1.26 | 1.93 | −0.93 | 2.58 | U | 255.50 | <0.0005 | ≤ - 1.67 | 12 | 1 |
| | Delayed Visual Recall | Delayed Recall (z) | 1.26 | 1.98 | −1.37 | 3.55 | U | 258.00 | <0.0005 | ≤ - 1.67 | 16 | 1 |
| | Visual Forgetting | D and P Visual Forgetting (z) | 0.33 | 0.00 | 0.33 | 1.83 | U | 433.00 | 0.0098 | ≤ - 1.67 | 8 | 1 |
| | Visual Recognition | Doors (z) | 0.67 | 1.33 | −0.67 | 1.75 | t | 3.74 | 0.0072 | ≤ - 1.67 | 7 | 1 |
| | | RMT Scenes (z) | 1.00 | 0.99 | −0.35 | 2.65 | U | 281.00 | 0.0025 | ≤ - 1.67 | 9 | 0 |
| | | Faces (z) | 0.00 | 2.33 | −0.33 | 1.66 | t | 1.29 | 0.6536 | ≤ - 1.67 | 8 | 6 |
| | Autobiographical Memory | AMI Childhood (9) | 9.00 | 3.00 | 5.00 | 4.00 | U | 174.50 | <0.0005 | ≤3.00 * | 11 | 1 |
| | | Early Adulthood (9) | 9.00 | 1.50 | 4.00 | 4.00 | U | 123.00 | <0.0005 | ≤3.00 * | 14 | 0 |
| Intelligence, Semantic Memory, and Language | Personal Semantic Memory | Childhood (21) | 19.50 | 3.00 | 18.00 | 5.00 | U | 267.00 | 0.0835 | ≤11.00 * | 3 | 0 |
| | | Early Adulthood (21) | 20.50 | 2.00 | 19.00 | 2.50 | U | 263.00 | 0.0687 | ≤14.00 * | 4 | 1 |
| | NART | p-FSIQ (z) | 1.44 | 0.85 | 1.04 | 1.05 | U | 486.00 | 0.1281 | ≤ - 1.67 | 0 | 0 |
| | WASI/ WASI-II | Vocabulary (z) | 1.40 | 1.25 | 0.70 | 1.20 | t | 3.05 | 0.0584 | ≤ - 1.67 | 0 | 0 |
| | | Similarities (z) | 1.05 | 0.80 | 0.70 | 0.85 | U | 378.00 | 0.1024 | ≤ - 1.67 | 0 | 0 |
| | GNT | (z) | 0.63 | 0.98 | 0.15 | 1.89 | U | 423.50 | 0.0683 | ≤ - 1.67 | 5 | 0 |
| | C and CT | (z) | 0.34 | 1.22 | 0.02 | 1.22 | U | 496.50 | 0.1484 | ≤ - 1.67 | 5 | 0 |

*Table 1 continued on next page*

Table 1 continued

| Domain | Test | Subtest | Controls | | Patients | | Controls vs patients | | | 'Impaired' range | | |
|---|---|---|---|---|---|---|---|---|---|---|---|---|
| | | | M | IQR | M | IQR | Test | Statistic | p-corr | cut-off score | Patients (n) | Controls (n) |
| Executive Function | WMS-III | Digit Span (z) | 0.84 | 1.25 | 0.33 | 1.67 | t | 2.70 | 0.1024 | ≤ - 1.67 | 2 | 1 |
| | DKEFS Trails | Number-Letter Switching (z) | 0.67 | 0.67 | 0.33 | 1.00 | U | 470.00 | 0.1024 | ≤ - 1.67 | 3 | 1 |
| Visuomotor Function | | Visual Scanning (z) | 0.67 | 1.50 | 0.00 | 1.34 | U | 584.00 | 0.6536 | ≤ - 1.67 | 5 | 4 |
| | | Motor Speed (z) | 0.67 | 1.00 | 0.33 | 1.34 | U | 552.00 | 0.4915 | ≤ - 1.67 | 7 | 5 |
| | ROCFT | copy rank | > 16th %ile | 0.00 | > 16th %ile | 0.00 | U | 619.00 | 0.4915 | ≤ 16th %ile | 2 | 1 |
| | VOSP | Cube Analysis (z) | 10.00 | 1.00 | 9.00 | 2.00 | U | 548.00 | 0.3116 | ≤6.00 ** | 3 | 0 |
| | | Dot Counting (z) | 10.00 | 0.00 | 10.00 | 0.00 | U | 655.50 | 0.6536 | ≤8.00 ** | 1 | 1 |
| | | Position Discrimination (z) | 20.00 | 0.00 | 20.00 | 1.00 | U | 673.00 | 0.6536 | ≤18.00 ** | 4 | 2 |
| Mood | HADS | Anxiety (21) | 4.00 | 4.00 | 5.00 | 5.50 | U | 420.00 | 0.0910 | ≥15.00 *** | 3 | 0 |
| | | Depression (21) | 1.00 | 1.00 | 3.00 | 4.50 | U | 298.00 | 0.0006 | ≥15.00 *** | 0 | 0 |

AMI: Autobiographical Memory Interview; D and P: Doors and People Test; DKEFS: Delis-Kaplan Executive Function System; GNT: Graded Naming Test; HADS: Hospital Anxiety and Depression Scale; IQR: Inter-Quartile Range; M: median; NART: National Adult Reading Test; p-corr: p values are corrected using the Holm-Bonferroni sequential correction for multiple comparisons (n = 35); RMT: Warrington Recognition Memory Tests (words, faces) and Warrington Topographical Memory test (scenes); ROCFT: Rey-Osterrieth Complex Figure Test; *t*: Student's t-test; *U*: Mann-Whitney U; VOSP: Visual Object and Space Perception Battery; WASI/WASI-II Wechsler Abbreviated Scale of Intelligence; WMS-III: Wechsler Memory Scale III; *Wt*: Welch's t-test; *,**,***: no standardized scores available for these subtests; *: highest score of 'definitely abnormal' range, that is scores at or below which none of the healthy controls scored in **Kopelman et al. (1989)**; **: 5% cut-off score; ***: cut-off score for severe range.

DOI: https://doi.org/10.7554/eLife.46156.003

that the effect of average HPC volume reduction on the rsALFF of the PCC was fully mediated by average thalamic volume reduction (**Figure 2—figure supplement 1**; **Figure 2—figure supplements 1—source data 1**).

## Resting-state functional connectivity

### Voxel-to-Voxel rsFC: connectome-MVPA

Capitalizing on the size of our patient cohort, we conducted a data-driven, whole-brain, principal components-based analysis, ('connectome-MVPA'), implemented in the Conn toolbox (**Whitfield-Gabrieli and Nieto-Castanon, 2012**). This method is used to identify the voxel clusters in which healthy controls and patient groups differ significantly with respect to their rsFC with the rest of the brain, instead of selecting seed/target regions or networks in an a priori fashion.

This analysis showed group differences in the whole-brain rsFC of a right HPC cluster (**Figure 2d**).

### Seed-to-Voxel rsFC

In order to identify the specific brain regions showing reduced rsFC with the right HPC in patients, the right HPC was then selected for a whole-brain seed-to-voxel rsFC analysis (**Biswal et al., 1995**; **Margulies et al., 2007**). The spatially unsmoothed timeseries data were extracted from participants' manually delineated right HPC in native space, ensuring that rsFC differences were not an artefact of insufficient co-registration of the atrophic HPC. This seed region showed reduced rsFC with clusters in the medial prefrontal and posteromedial (PCC, retrosplenial, and precuneus) cortices, and the left HPC (**Figure 2e–g**). We also wanted to ensure that the left HPC cluster that showed reduced rsFC with the right HPC was not a result of suboptimal co-registration of the functional images with

**Table 2.** Clinical details of autoimmune LE patients (acute phase).

| Code | Age (years) | Sex | Antibody type | Acute T2 scan notes HPC R | L | Other structures |
|---|---|---|---|---|---|---|
| 1 | 65.75 | M | LGI1 | Normal T2 signal and volume; facilitated diffusion | High T2 signal; swelling; normal diffusion | L AMG: high T2 signal |
| 2 | 69.98 | F | VGKCC | Normal T2 signal; mild atrophy; facilitated diffusion | High T2 signal; normal volume; facilitated diffusion | No abnormalities |
| 3 | 62.23 | M | VGKCC | Normal T2 signal and volume; facilitated diffusion | High T2 signal; swelling; normal diffusion | L AMG, L ERC: high T2 signal |
| 4 | 46.41 | M | LGI1 | High T2 signal; normal volume; normal diffusion | Normal T2 signal and volume; facilitated diffusion | No abnormalities |
| 5 | 56.65 | M | LGI1 | L/R: high T2 signal | | No abnormalities |
| 6 | 58.18 | M | LGI1 | No abnormalities | | |
| 7 | 56.13 | M | LGI1 | Normal volume and signal | High T2 signal; swelling | L/R AMG: high T2 signal |
| 8 | 76.54 | M | LGI1 | High T2 signal; normal volume and diffusion | High T2 signal; normal volume; facilitated diffusion | No abnormalities |
| 9 | 54.94 | M | LGI1 | High T2 signal; swelling; normal diffusion | High T2 signal; swelling; normal diffusion | No abnormalities |
| 10 | 44.81 | M | LGI1 | L/R: high T2 signal; swelling | | No abnormalities |
| 11 | 45.77 | M | LGI1 | High T2 signal; normal volume | High T2 signal; normal volume | No abnormalities |
| 12 | 46.06 | M | LGI1/Caspr2 | High T2 signal; atrophy | Normal T2 signal and volume | No abnormalities |
| 13 | 35.75 | M | LGI1/Caspr2 | L/R: normal T2 signal; mild atrophy; normal diffusion | | No abnormalities |
| 14 | 72.08 | M | LGI1 | High T2 signal; mild atrophy; facilitated diffusion | Normal T2 signal; atrophy; facilitated diffusion | No abnormalities |
| 15 | 52.28 | M | LGI1 | High T2 signal; normal volume; facilitated diffusion | Normal T2 signal and volume; facilitated diffusion | No abnormalities |
| 16 | 52.48 | M | LGI1/Caspr2 | High T2 signal; swelling; facilitated diffusion | Normal T2 signal and volume; facilitated diffusion | No abnormalities |
| 17 | 51.62 | M | VGKCC | High T2 signal; swelling | Normal T2 signal and volume | No abnormalities |
| 18 | 75.18 | M | LGI1 | L/R: high T2 signal; swelling; normal diffusion | | L/R AMG: high T2 signal; swelling |
| 19 | 78.73 | M | LG1/Caspr2 | High T2 signal; mild atrophy; normal diffusion | High T2 signal; normal volume; normal diffusion | No abnormalities |
| 20 | 53.75 | F | LGI1 | L/R: high T2 signal; normal volume and diffusion | | No abnormalities |
| 21 | 73.68 | F | VGKCC | L/R: high T2 signal; swelling; facilitated diffusion | | No abnormalities |
| 22 | 63.59 | M | LGI1 | L/R: high T2 signal; normal volume and diffusion | | No abnormalities |
| 23 | 60.35 | M | VGKCC | No abnormalities | | |
| 24 | 54.30 | M | VGKCC | L/R: high T2 signal; atrophy | | L/R AMG: high T2 signal; atrophy |
| 25 | 52.70 | M | seronegative | L/R: high T2 signal | | No abnormalities |
| 26 | 47.43 | F | seronegative | No abnormalities | | |
| 27 | 58.60 | M | seronegative | L/R: high T2 signal | | No abnormalities |
| 28 | 25.42 | M | Anti-Ma2 | L/R: high T2 signal and swelling | | No abnormalities |
| 29 | 45.77 | F | seronegative | L/R: high T2 signal | | No abnormalities |
| 30 | 16.64 | F | GAD | No abnormalities | | |
| 31 | 71.35 | M | seronegative | L/R: high T2 signal; atrophy | | No abnormalities |
| 32 | 60.44 | M | VGKCC | L/R: atrophy | | PHC atrophy |
| 33 | 53.48 | M | seronegative | L/R: atrophy | | No abnormalities |
| 34 | 64.87 | F | seronegative | L/R: atrophy | | No abnormalities |
| 35 | 47.32 | F | seronegative | L/R: high T2 signal | | R AMG: high T2 signal; swelling |

*Table 2 continued on next page*

*Table 2 continued*

| Code | Age (years) | Sex | Antibody type | Acute T2 scan notes | | Other structures |
|------|-------------|-----|---------------|---------------------|---|------------------|
| | | | | **HPC** | | |
| | | | | **R** | **L** | |
| 36 | 61.88 | F | seronegative | L/R: high T2 signal; atrophy | | No abnormalities |
| 37 | 71.90 | F | seronegative | L/R: high T2 signal (especially R) | | No abnormalities |
| 38 | 34.49 | F | GAD | L/R: high T2 signal | | No abnormalities |

Age: age at symptom onset (years); AMG: Amygdala; Caspr2: anti-contactin-associated protein-like 2; ERC: entorhinal cortex; F = female; GAD: anti-gluta-mic acid decarboxylase autoantibody; HPC: hippocampus; L: left hemisphere; LGI1: anti-leucine-rich glioma-inactivated1; M = male; PHC: parahippocam-pal cortex; R: right hemisphere; VGKCC: anti-voltage-gated potassium channel complex. The clinical details of patients 1–24 have also been presented in *Loane et al. (2019)*.

DOI: https://doi.org/10.7554/eLife.46156.022

**Table 3.** Volumetry of MTL and subcortical structures in autoimmune LE patients.

| Structure | Controls | | Patients | | Mean % reduction | $F$ | Partial $\eta^2$ | p-corr |
|-----------|----------|---|----------|---|------------------|-----|------------------|--------|
| | Mean (mm$^3$) | SD (mm$^3$) | Mean (mm$^3$) | SD (mm$^3$) | | | | |
| R HPC | 3648.99 | 459.59 | 2733.87 | 751.09 | −25.08 | 63.38 | 0.390 | <0.0005 |
| L HPC | 3439.48 | 431.91 | 2671.18 | 710.65 | −22.34 | 46.02 | 0.317 | <0.0005 |
| R ERC | 1602.69 | 324.07 | 1254.97 | 404.70 | −21.70 | 10.76 | 0.119 | 0.0308 |
| L ERC | 1508.83 | 326.07 | 1200.18 | 432.07 | −20.46 | 9.48 | 0.106 | 0.0534 |
| R Thalamus | 7407.51 | 762.15 | 7072.79 | 845.09 | −4.52 | 8.23 | 0.077 | 0.0900 |
| L Thalamus | 7633.09 | 788.11 | 7194.76 | 797.73 | −5.74 | 15.41 | 0.135 | 0.0034 |
| R PRC | 1791.42 | 378.97 | 1561.61 | 403.86 | −12.83 | 7.11 | 0.082 | 0.1575 |
| L PRC | 1812.50 | 523.31 | 1601.39 | 478.78 | −11.65 | 2.88 | 0.035 | >0.9999 |
| R PHC | 1900.73 | 423.85 | 1665.68 | 331.28 | −12.37 | 6.36 | 0.074 | 0.2240 |
| L PHC | 2016.52 | 435.92 | 1851.53 | 445.52 | −8.18 | 1.53 | 0.019 | >0.9999 |
| R AMG | 1395.67 | 267.38 | 1313.76 | 406.51 | −5.87 | 5.15 | 0.060 | 0.3899 |
| L AMG | 1321.96 | 217.40 | 1268.47 | 384.22 | −4.05 | 4.43 | 0.053 | 0.5320 |
| R Nacc | 339.42 | 107.77 | 318.53 | 109.69 | −6.16 | 0.25 | 0.003 | >0.9999 |
| L Nacc | 433.79 | 127.41 | 381.89 | 153.66 | −11.96 | 3.70 | 0.036 | 0.7410 |
| R TPC | 4558.60 | 846.26 | 4520.16 | 1026.08 | −0.84 | 0.12 | 0.002 | >0.9999 |
| L TPC | 4331.40 | 742.35 | 4512.95 | 795.76 | 4.19 | 1.90 | 0.023 | >0.9999 |
| R Putamen | 4332.54 | 548.04 | 4157.00 | 630.21 | −4.05 | 0.85 | 0.008 | >0.9999 |
| L Putamen | 4382.39 | 705.40 | 4151.89 | 662.93 | −5.26 | 1.67 | 0.017 | >0.9999 |
| R Caudate | 3403.96 | 440.42 | 3369.16 | 453.63 | −1.02 | 0.06 | 0.001 | >0.9999 |
| L Caudate | 3211.40 | 438.40 | 3134.26 | 497.42 | −2.40 | 0.12 | 0.001 | >0.9999 |
| R Pallidum | 1705.61 | 260.63 | 1630.00 | 251.08 | −4.43 | 0.90 | 0.009 | >0.9999 |
| L Pallidum | 1720.40 | 297.86 | 1661.13 | 308.18 | −3.45 | 0.41 | 0.004 | >0.9999 |
| brainstem | 22119.06 | 2258.25 | 21931.42 | 2647.93 | −0.85 | 0.23 | 0.002 | >0.9999 |

Volumetry of manually and automatically delineated MTL and other subcortical structures of all patients (n = 38). Volumes for each structure are compared between patients and controls, using age, sex, TIV, and scan source (MAP, OPTIMA) as between-subjects covariates in a series of univariate ANCOVAs; AMG: amygdala; ANCOVA: analysis of covariance; ; ERC: entorhinal cortex; HPC: hippocampus; L: left hemisphere; MAP: Memory and Amnesia Project; MTL: medial temporal lobe; Nacc: nucleus accumbens; OPTIMA: Oxford Project To Investigate Memory and Aging; p-corr: p values are adjusted with the Holm-Bonferroni sequential correction method for multiple comparisons (n = 23); PHC: parahippocampal cortex; PRC: perirhinal cortex; R: right hemisphere; SD: standard deviation; TIV: total intracranial volume; TPC: temporopolar cortex.

DOI: https://doi.org/10.7554/eLife.46156.004

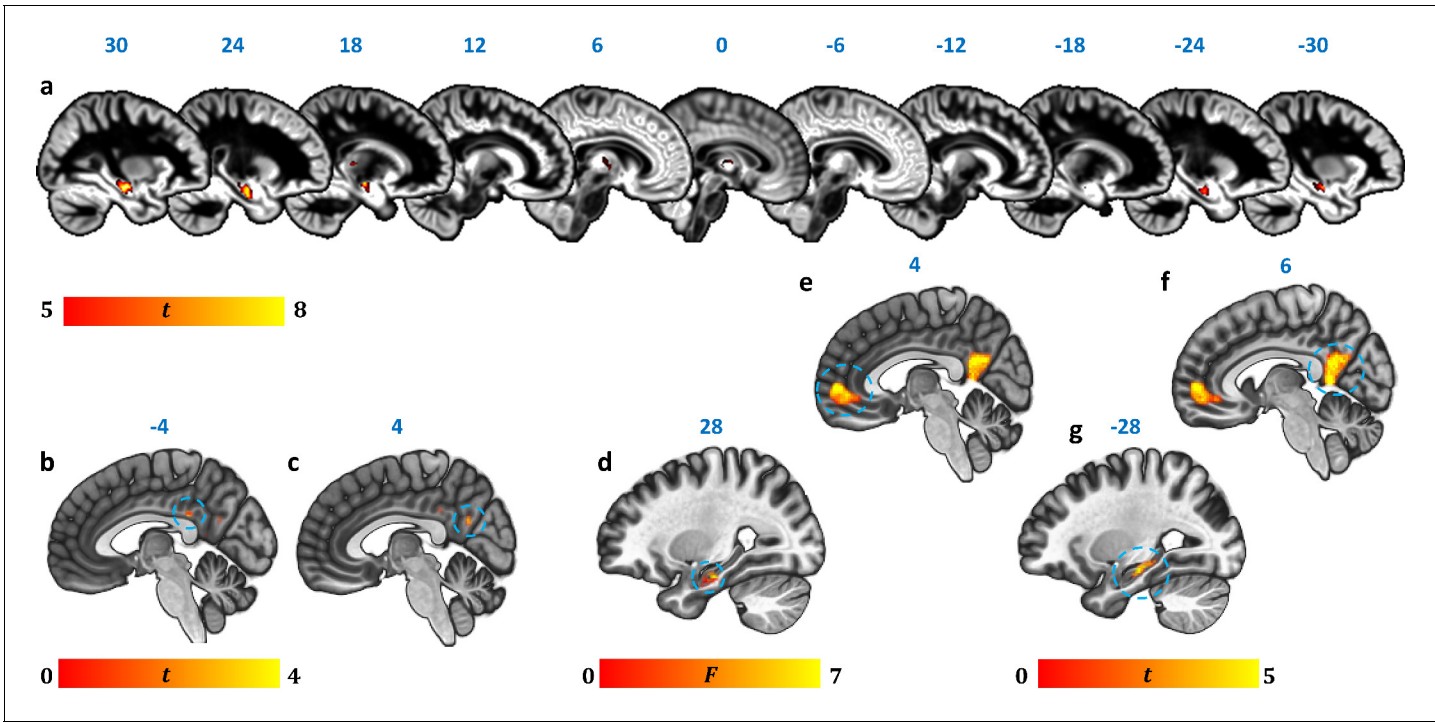

**Figure 2.** Reduction in GM volume, rsALFF and rsFC in autoimmune LE patients. (a) A whole-brain VBM on GM volume (contrast: controls > patients) showed volume reduction in patients' HPC bilaterally, as well as in mediodorsal-anterior and right dorsolateral thalamic regions (*Table 4*); clusters survive FWE peak-level correction (p<0.05) over p<0.001 unc; color bar indicates t values; b-c: Reduced rsALFF in patients in (b) the posterior cingulate (kE = 89, p-FWE = 0.033; peak voxel coordinates: −4,−36, 28) and (c) precuneus (kE = 137; p-FWE = 0.003; peak voxel coordinates: 4,−60, 26); d-j: reduced rsFC in patients; d: a whole-brain MVPA (omnibus F) showed abnormal rsFC for patients in a cluster in the right HPC (kE = 178, p-FWE = 0.001; peak voxel coordinates: 28,−16,−20; color bar indicates F values); e-g: reduced rsFC of the right HPC (whole-brain seed-to-voxel analysis; seed: right HPC, anatomically delineated in native space, unsmoothed timeseries; contrast: controls > patients); e: medial prefrontal cortex (kE = 1152, p-FWE <0.0001, peak voxel coordinates: 4, 56, 2); f: posteromedial cortex (posterior cingulate, retrosplenial cortex, precuneus; kE = 986, p-FWE <0.0001, peak voxel coordinates: 6,−50, 8); g: left HPC (kE = 393, p-FWE <0.0001, peak voxel coordinates: −12,−36, 2). All rsFC and rsALFF clusters survive FWE correction (p<0.05) for cluster size over an individual voxel threshold of p<0.001; FWE: family-wise error; HPC: hippocampus; kE: cluster size (number of voxels); rsALFF: Resting-state amplitude of low frequency fluctuations; rsFC: Resting-state functional connectivity; VBM: voxel-based morphometry.

DOI: https://doi.org/10.7554/eLife.46156.005

The following source data and figure supplements are available for figure 2:

**Figure supplement 1.** Relationship of HPC atrophy with PCC functional abnormalities.

DOI: https://doi.org/10.7554/eLife.46156.006

**Figure supplement 1—source data 1.** This spreadsheet contains the mean GM volume of the HPC and thalamic VBM clusters and the mean rsALFF in the PCC cluster (z-res) for healthy controls and patients that are plotted in *Figure 2—figure supplement 1*; GM: gray matter; HPC: hippocampus; MAP: Memory and Amnesia Project; OPTIMA: Oxford Project To Investigate Memory and Aging; PCC: posterior cingulate cortex; rsALFF: resting-state amplitude of low frequency fluctuations; TIV: total intracranial volume; VBM: voxel-based morphometry; z-res: GM volumes from VBM clusters are residualized against age, sex, scan source (MAP, OPTIMA), and TIV across participants; mean rsALFF is residualized against age and sex across participants.

DOI: https://doi.org/10.7554/eLife.46156.007

the atrophic HPC. For further correlational analyses (section below), we thus used the mean rsFC values between the right and left HPC in native space (unsmoothed time series) instead of the left HPC cluster. Patients' reduced inter-HPC rsFC (of all 13 structural and functional abnormalities) negatively correlated, across patients, with the delay between the onset of their symptoms and the time they underwent our research MRI (Supplementary Table 2 in *Supplementary file 1*).

## Structure/Function-Behavior Correlations

Having identified the core brain abnormalities in our patient group, we investigated the contributions of these to explaining memory impairment.

**Table 4.** GM volume reduction in autoimmune LE patients (whole-brain VBM).

| | Peak | | | | | Center of mass | | | |
|---|---|---|---|---|---|---|---|---|---|
| kE | p-FWE | T | X | Y | Z | X | Y | Z | Structure |
| 2574 | <0.0005 | 7.53 | 28 | −17 | −20 | 28 | −16 | −18 | R HPC |
| 910 | <0.0005 | 6.81 | −29 | −12 | −18 | −27 | −15 | −19 | L HPC |
| 113 | 0.002 | 6.18 | 19 | −28 | 5 | 18 | −26 | 6 | R lateral thalamus |
| 414 | 0.006 | 5.85 | -1 | −16 | 0 | 3 | −12 | 7 | anterior/mediodorsal thalamus |

Contrast: controls > patients; covariates: age, sex, scan source (MAP, OPTIMA), and TIV. Clusters are FWE-corrected at peak-voxel level (p<0.05) over an individual voxel threshold of p<0.001 (unc.); voxel size: 1 mm$^3$ isotropic; spatial smoothing kernel: 4 mm FWHM; FWHM: Full-width at half-maximum; HPC: hippocampus; kE: cluster size (number of voxels; minimum cluster size: 50 voxels; L: Left hemisphere; mm: millimeter; MAP: Memory and Amnesia Project; OPTIMA: Oxford Project To Investigate Memory and Aging; R: Right hemisphere; TIV: total intracranial volume; x, y, z: coordinates in mm.

DOI: https://doi.org/10.7554/eLife.46156.008

We first applied a stringent correction for multiple testing for the total number of correlations conducted (n = 39) between the brain abnormalities identified (n = 13; right/left HPC volumes and right entorhinal cortical volumes, based on manual delineation; left thalamic volumes, based on automated segmentation; anterior-mediodorsal and right dorsolateral thalamic volumes and right/left HPC volumes, expressed by the VBM clusters above; reduced rsALFF in the precuneus and the PCC; reduced rsFC between the right HPC and the left HPC, the medial prefrontal cortex, and the precuneus) and the three composite memory scores (n = 3; anterograde retrieval; anterograde retention; remote autobiographical memory). Three correlations survived correction for multiple tests: i) Anterograde retrieval scores correlated across patients with their reduced rsALFF in the PCC (r = 0.551; p-corr = 0.024); ii) Anterograde retention (i.e. 'forgetting') scores correlated with patients' reduced right HPC volume (VBM cluster; rho = 0.556, p-corr = 0.024; manually delineated right HPC volume: rho = 0.508, p-corr = 0.079); iii) Remote autobiographical memory scores correlated with patients' reduced volume in the left thalamus (r = 0.558; p-corr = 0.041; rest of ps, p-corr ≥0.105; Supplementary Table 3 in *Supplementary file 1*).

Given the striking lack of correlations with HPC volume, we addressed the possibility of false negatives in our original approach by iterating the correlational analyses above after introducing three amendments: i) we fragmented the anterograde retrieval composite score into four composite scores (visual/verbal recall/recognition), taking into account the possibility of different relationships of recall vs. recognition memory scores with brain abnormalities; ii) we applied a more lenient correction for the number of structural/functional abnormalities (n = 13), separately for each composite score examined (n = 6; visual/verbal recall/recognition, remote autobiographical memory, visual forgetting; Supplementary Table 4 in *Supplementary file 1*); iii) in a post-hoc fashion, we examined the relationship of these memory scores with the manually delineated anterior vs. posterior HPC portions at uncorrected levels (Supplementary Table 5 in *Supplementary file 1*).

### Anterograde memory: Verbal recognition

Verbal recognition scores correlated with patients' reduced inter-HPC rsFC (r = 0.498, p-corr = 0.039; rest of ps, p-corr ≥0.18; *Figure 3b*; *Figure 3—source data 1*). Out of the HPC volumes, only the volume expressed by the left HPC VBM cluster correlated with verbal recognition scores at uncorrected levels (r = 0.360, p-unc = 0.029; *Figure 3a*; rest of rs, 0.281 ≥ r ≥ 0.276; rest of ps, 0.098 ≥ p unc ≥ 0.092). None of the HPC volumes examined (left/right HPC VBM clusters; manually delineated left/right HPC) correlated with inter-HPC rsFC across patients (all rs, |r| ≤ 0.135; all ps, p-unc ≥ 0.441). We thus entered mean inter-HPC rsFC and one of those four HPC volume measures as independent variables in four separate multiple step-wise linear regressions, with verbal recognition scores as the dependent variable. In all four analyses, the regression terminated in a single step, with inter-HPC rsFC as the only predictor of patients' performance (R$^2$ = 0.25; β(z) = 0.50; F = 10.53, p = 0.003).

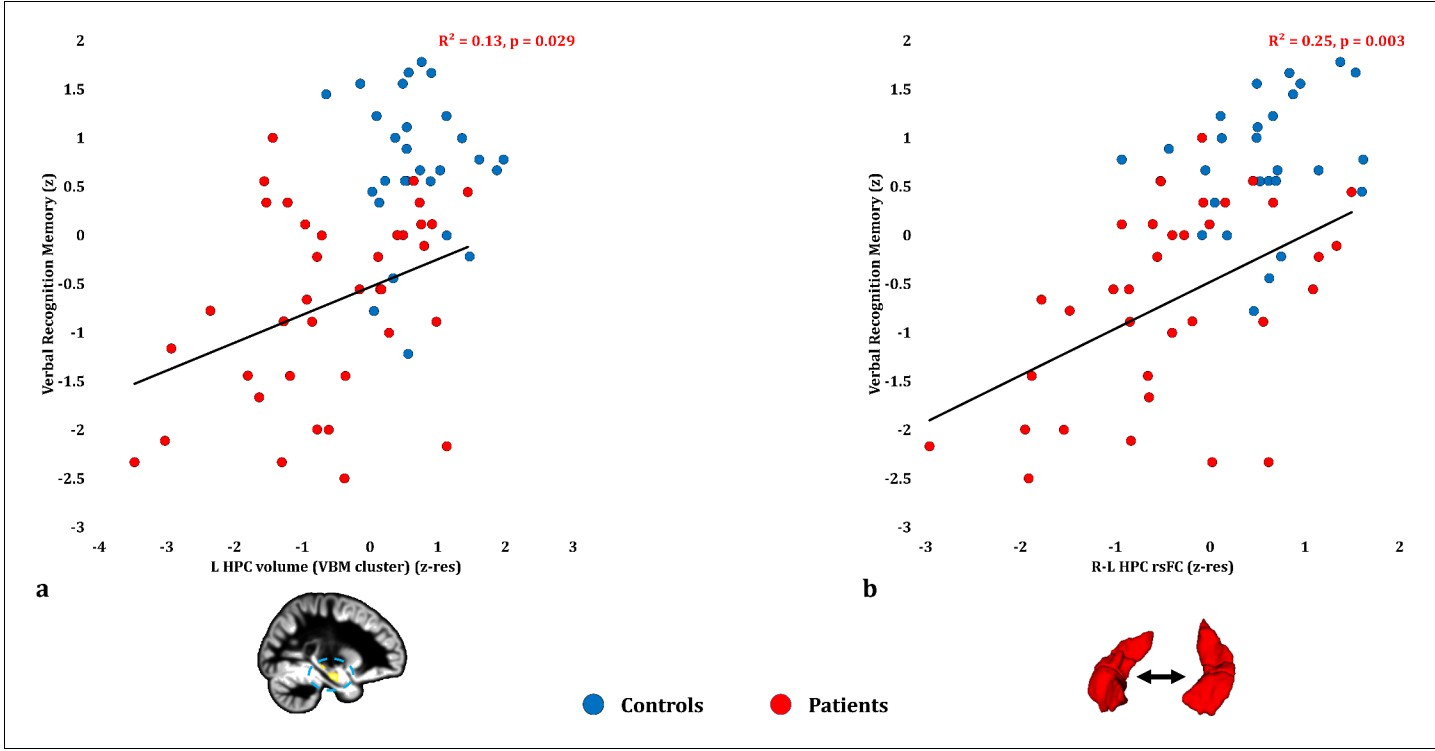

**Figure 3.** Verbal Recognition Memory: Structural/Functional correlates. (a) GM volume expressed by the left HPC VBM cluster correlated across patients with their verbal recognition composite memory scores only at uncorrected levels (r = 0.36, p-unc = 0.029); (b) mean inter-HPC rsFC correlated across patients with their verbal recognition composite memory scores and survived correction for multiple (structural/functional brain abnormalities examined: n = 13) testing (r = 0.50, p-corr = 0.039); GM: gray matter; HPC: hippocampus; L, R: left, right (hemisphere); MAP: Memory and Amnesia Project; OPTIMA: Oxford Project To Investigate Memory and Aging; p: significance values are presented at uncorrected levels; rho: Spearmann's rank correlation coefficient; rsFC: resting-state functional connectivity; VBM: voxel-based morphometry; z: memory scores are averaged age-scaled and standardized scores of participants' performance in the subtests of interest; z-res: GM volume from VBM clusters is residualized against age, sex, scan source (MAP, OPTIMA) and TIV across participants; mean rsFC is residualized across participants against age and sex.
DOI: https://doi.org/10.7554/eLife.46156.009

The following source data is available for figure 3:

**Source data 1.** This spreadsheet contains the mean GM volume of the left HPC VBM cluster and the mean inter-HPC rsFC (z-res) and the verbal recognition memory composite scores (z) of healthy controls and patients that are plotted in *Figure 3*; GM: gray matter; HPC: hippocampus; MAP: Memory and Amnesia Project; OPTIMA: Oxford Project To Investigate Memory and Aging; rsFC: resting-state functional connectivity; VBM: voxel-based morphometry; z: memory scores are averaged age-scaled and standardized scores of participants' performance in the subtests of interest; z-res: GM volumes from VBM clusters are residualized against age, sex, scan source (MAP, OPTIMA), and TIV across participants; mean rsFC is residualized against age and sex across participants. These data can be opened with Microsoft Excel or with open-source alternatives such as OpenOffice.
DOI: https://doi.org/10.7554/eLife.46156.010

### Anterograde memory: Visual recognition

Visual recognition scores correlated with patients' reduced rsALFF in the PCC (r = 0.543, p-corr = 0.014), and only marginally with the volume of the left HPC VBM cluster (r = 0.449, p-corr = 0.072; rest of ps, p-corr ≥0.110). The volume of the left HPC VBM cluster correlated with the rsALFF in the PCC across patients (r = 0.449, p=0.007).

Patients' reduced rsALFF in the PCC correlated with visual recognition memory scores over and above the correlative reduction in the left HPC volume (VBM cluster) (partial correlation: r = 0.432, p=0.013). Since left HPC volumes correlated across patients with their reduced rsALFF in the PCC, and since both these abnormalities correlated with patients' impaired visual recognition memory, we conducted a mediation analysis to test our hypothesis that the effects of HPC damage trigger abnormalities within the extended HPC system, and that these underlie memory impairment (see Materials and methods section). Indeed, the effects of left HPC volume reduction on visual recognition scores were fully mediated by the reduction in PCC rsALFF (direct effect: β = 0.24, p=0.183;

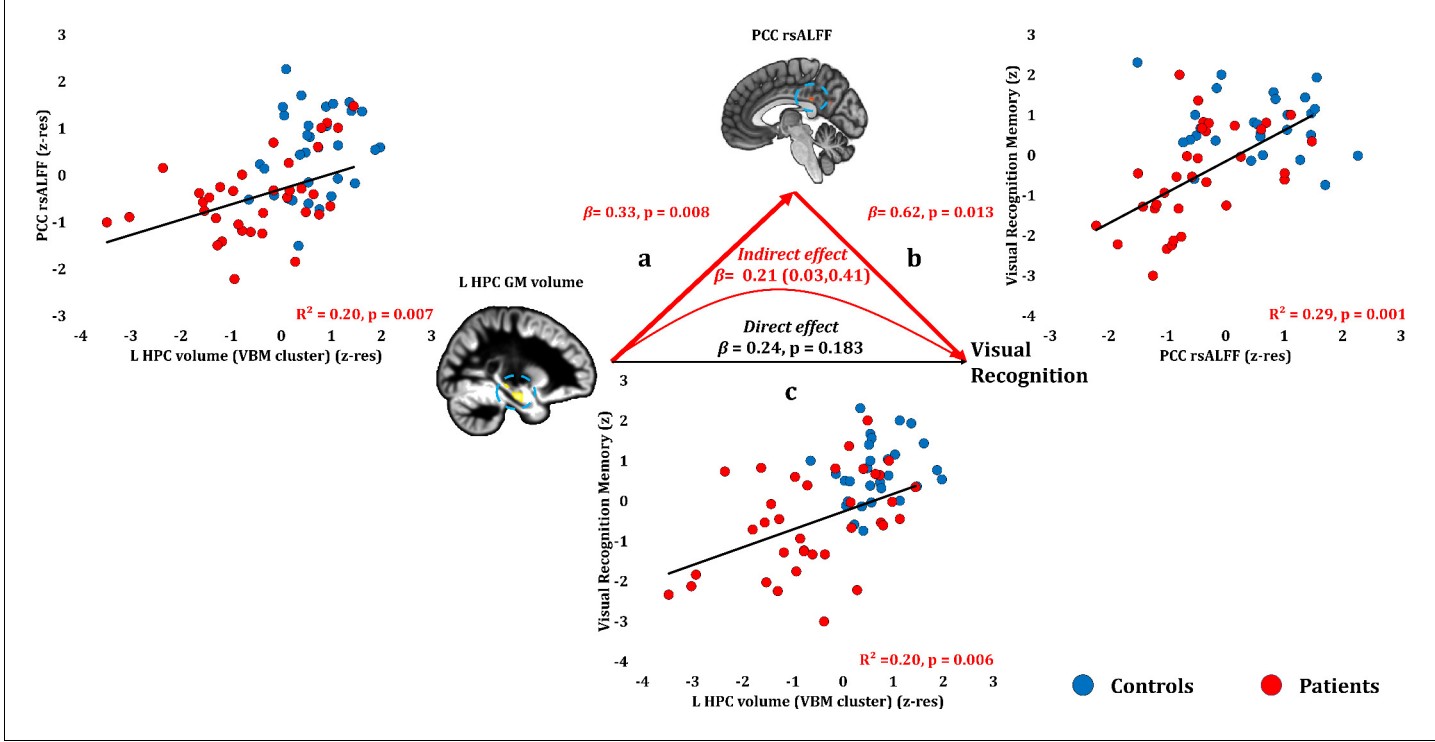

**Figure 4.** Visual Recognition Memory: Structural/Functional correlates. (**a**) mean GM volume of the left HPC cluster correlated with the mean rsALFF of the PCC cluster across patients; (**b**) visual recognition memory scores correlated across patients with their mean rsALFF in the PCC cluster, surviving correction for multiple testing for the 13 structural/functional abnormalities examined (r = 0.54, p-corr = 0.014); the mediation analysis demonstrates that this effect held over and above the correlation of PCC rsALFF with the mean GM volume of the left HPC cluster; (**c**) mean GM volume of the left HPC cluster correlated with visual recognition memory scores across patients, but did not survive correction for multiple testing (r = 0.45, p-corr = 0.072); however, the mediation analysis demonstrated that this relationship did not hold over and above the correlation of the mean GM volume of the left HPC cluster with the mean PCC rsALFF; there was only an indirect effect of reduced HPC GM volume on visual recognition memory (within parenthesis: 95% confidence intervals); GM: gray matter; HPC: hippocampus; L: left (hemisphere); MAP: Memory and Amnesia Project; OPTIMA: Oxford Project To Investigate Memory and Aging; p: significance values are presented at uncorrected levels; PCC: posterior cingulate cortex; rsALFF: resting-state amplitude of low frequency fluctuations; TIV: total intracranial volume; VBM: voxel-based morphometry; z: memory scores are averaged age-scaled and standardized scores of participants' performance in the subtests of interest; z-res: GM volumes from VBM clusters are residualized against age, sex, scan source (MAP, OPTIMA), and TIV across participants; mean rsALFF values are residualized across participants against age and sex.
DOI: https://doi.org/10.7554/eLife.46156.011

The following source data is available for figure 4:

**Source data 1.** This spreadsheet contains the mean GM volume of the left HPC VBM cluster and the mean rsALFF in the PCC (z-res) and the visual recognition memory composite scores (z) of healthy controls and patients that are plotted in *Figure 4*; GM: gray matter; HPC: hippocampus; MAP: Memory and Amnesia Project; OPTIMA: Oxford Project To Investigate Memory and Aging; PCC: posterior cingulate cortex; rsALFF: resting-state amplitude of low-frequency fluctuations; VBM: voxel-based morphometry; z: memory scores are averaged age-scaled and standardized scores of participants' performance in the subtests of interest; z-res: GM volumes from VBM clusters are residualized against age, sex, scan source (MAP, OPTIMA), and TIV across participants; mean rsALFF is residualized against age and sex across participants. These data can be opened with Microsoft Excel or with open-source alternatives such as OpenOffice.
DOI: https://doi.org/10.7554/eLife.46156.012

indirect effect: β = 0.21, 95% CI: 0.03, 0.41) (*Figure 4*; *Figure 4—source data 1*). In other words, controlling for the mediator variable (rsALFF in the PCC) reduced the variance of the dependent variable (scores of visual recognition memory) explained by the independent variable (volume expressed by the left HPC VBM cluster), to the extent that the relationship between the independent and dependent variables became non-significant.

## Anterograde memory: Verbal recall

Verbal recall memory composite scores correlated across patients with their reduced PCC rsALFF (r = 0.582, p-corr = 0.004) and with the volume of the left HPC VBM cluster (r = 0.495,

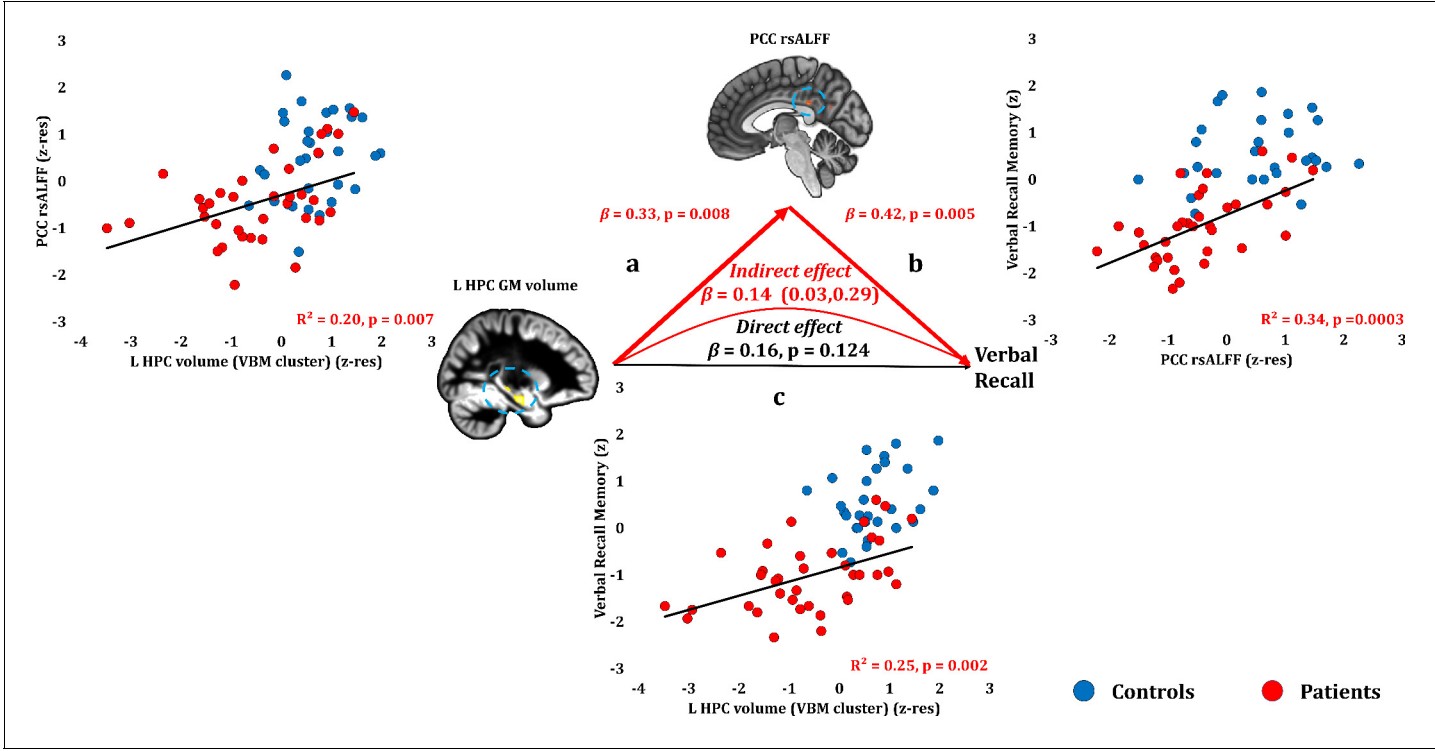

**Figure 5.** Verbal Recall Memory: Structural/Functional correlates. (a) mean GM volume of the left HPC cluster correlated with the mean rsALFF of the PCC cluster across patients; (b) verbal recall memory scores correlated across patients with their mean rsALFF in the PCC cluster, surviving corrections for multiple testing across the 13 structural/functional abnormalities examined (r = 0.582, p-corr = 0.004); the mediation analysis demonstrates that this effect held over and above the correlation of PCC rsALFF with the mean GM volume of the left HPC cluster; (c) mean GM volume of the left HPC cluster correlated with verbal recall memory scores across patients (r = 0.495, p-corr = 0.024); however, the mediation analysis demonstrated that this relationship did not hold over and above the correlation of the mean GM volume of the left HPC cluster with the mean PCC rsALFF; there was only an indirect effect of reduced HPC GM volume on verbal recall memory (within parenthesis: 95% confidence intervals); GM: gray matter; HPC: hippocampus; L: left (hemisphere); MAP: Memory and Amnesia Project; OPTIMA: Oxford Project To Investigate Memory and Aging; PCC: posterior cingulate cortex; rsALFF: resting-state amplitude of low frequency fluctuations; TIV: total intracranial volume; VBM: voxel-based morphometry; z: memory scores are averaged age-scaled and standardized scores of participants' performance in the subtests of interest; z-res: GM volumes from VBM clusters are residualized against age, sex, scan source (MAP, OPTIMA), and TIV across participants; mean rsALFF values are residualized across participants against age and sex.

DOI: https://doi.org/10.7554/eLife.46156.013

The following source data is available for figure 5:

**Source data 1.** This spreadsheet contains the mean GM volume of the left HPC VBM cluster and the mean rsALFF in the PCC (z-res) and the verbal recall memory composite scores (z) of healthy controls and patients that are plotted in *Figure 5*; GM: gray matter; HPC: hippocampus; MAP: Memory and Amnesia Project; OPTIMA: Oxford Project To Investigate Memory and Aging; PCC: posterior cingulate cortex; rsALFF: resting-state amplitude of low-frequency fluctuations; VBM: voxel-based morphometry; z: memory scores are averaged age-scaled and standardized scores of participants' performance in the subtests of interest; z-res: GM volumes from VBM clusters are residualized against age, sex, scan source (MAP, OPTIMA), and TIV across participants; mean rsALFF is residualized against age and sex across participants. These data can be opened with Microsoft Excel or with open-source alternatives such as OpenOffice.

DOI: https://doi.org/10.7554/eLife.46156.014

p-corr = 0.024; rest of ps, p-corr ≥0.290). Patients' reduced rsALFF in the PCC correlated with verbal recall scores over and above the left HPC volume reduction (VBM cluster) (partial correlation: r = 0.474, p=0.005). Consistent with our hypothesis, the effects of HPC atrophy on verbal recall were fully mediated by the reduction in PCC rsALFF (direct effect: β = 0.16, p=0.124; indirect effect: β = 0.14, 95% CI: 0.03, 0.29; *Figure 5*; *Figure 5—source data 1*).

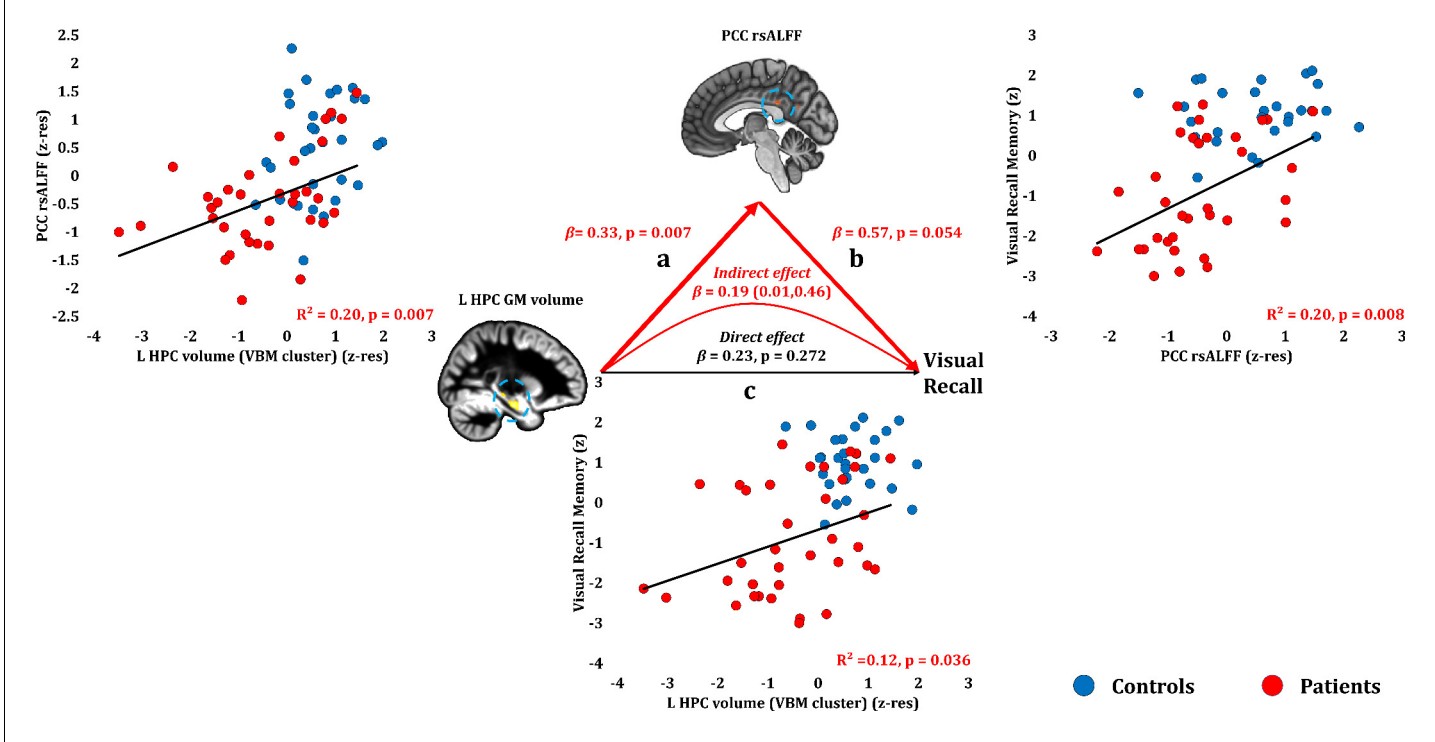

**Figure 6.** Visual Recall Memory: Structural/Functional correlates. (**a**) mean GM volume of the left HPC cluster correlated with the mean rsALFF of the PCC cluster across patients; (**b**) visual recall memory scores correlated at uncorrected levels across patients with their mean rsALFF in the PCC cluster; the mediation analysis demonstrates that this effect held over and above the correlation of PCC rsALFF with the mean GM volume of the left HPC cluster; (**c**) mean GM volume of the left HPC cluster correlated with visual recall memory scores across patients at uncorrected levels; however, the mediation analysis demonstrated that this relationship did not hold over and above the correlation of the mean GM volume of the left HPC cluster with the mean PCC rsALFF; there was only an indirect effect of reduced HPC GM volume on visual recall memory (within parenthesis: 95% confidence intervals); GM: gray matter; HPC: hippocampus; L: left (hemisphere); MAP: Memory and Amnesia Project; OPTIMA: Oxford Project To Investigate Memory and Aging; p: significance values are presented at uncorrected levels; PCC: posterior cingulate cortex; rsALFF: resting-state amplitude of low frequency fluctuations; TIV: total intracranial volume; VBM: voxel-based morphometry; z: memory scores are averaged age-scaled and standardized scores of participants' performance in the subtests of interest; z-res: GM volumes from VBM clusters are residualized against age, sex, scan source (MAP, OPTIMA), and TIV across participants; mean rsALFF values are residualized across participants against age and sex.

DOI: https://doi.org/10.7554/eLife.46156.015

The following source data is available for figure 6:

**Source data 1.** This spreadsheet contains the mean.

DOI: https://doi.org/10.7554/eLife.46156.016

## Anterograde memory: Visual recall

Visual recall scores correlated at uncorrected levels with PCC rsALFF (rho = 0.446, p-unc = 0.008; p-corr = 0.107), and the left HPC volume (VBM cluster) (rho = 0.370, p-unc = 0.026; p-corr = 0.312; rest of ps, p-corr ≥0.594). Across patients, the reduced rsALFF in the PCC marginally correlated with visual recall memory scores over and above their reduced left HPC volume (VBM cluster) (partial correlation: rho = 0.33, p=0.058). The effects of HPC atrophy on visual recall were fully mediated by the reduction in PCC rsALFF (direct effect: β = 0.23, p=0.272; indirect effect: β = 0.19, 95% CI:0.01,0.46) (*Figure 6*; *Figure 6—source data 1*).

## Anterograde memory: Retention (Forgetting)

Anterograde retention (visual forgetting) scores correlated only with volume reduction in the right HPC VBM cluster (rho = 0.556, p-corr = 0.008) and in the manually delineated right HPC (rho = 0.508, p-corr = 0.026; an alternative analysis, comparing patients that scored at ceiling with the rest of the patient group, is reported in Supplementary Table 6 in *Supplementary file 1*, disclosing the same relationships). No extra-HPC abnormalities correlated significantly with visual

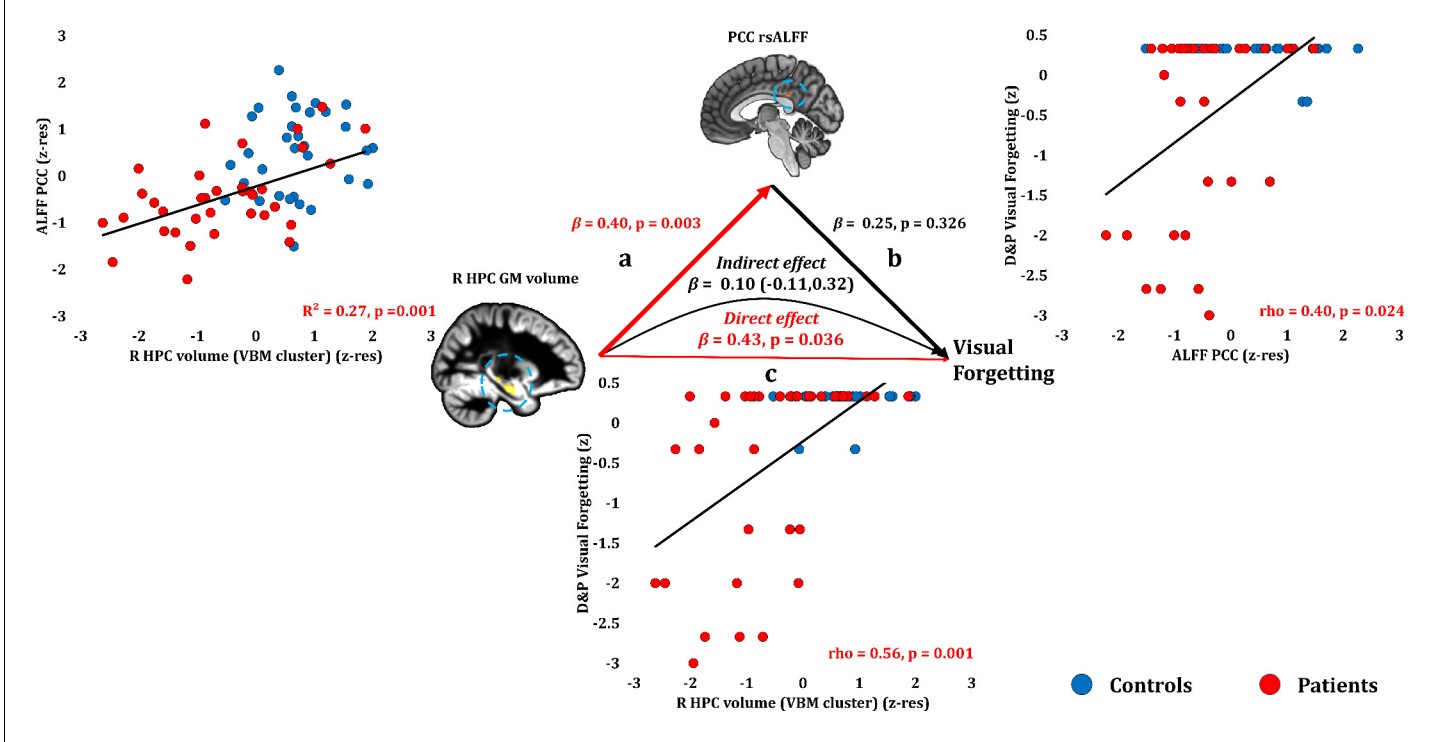

**Figure 7.** Forgetting: Structural/Functional correlates. (**a**) mean GM volume of the right HPC cluster correlated with the mean rsALFF of the PCC cluster across patients; (**b**) visual forgetting scores correlated across patients with their mean rsALFF in the PCC cluster, without, however, surviving correction for multiple tests (rho = 0.40, p-corr = 0.216); the mediation analysis demonstrates that this effect did not hold when the correlation of PCC rsALFF with the mean GM volume of the right HPC cluster was accounted for; (**c**) mean GM volume of the right HPC cluster correlated with visual forgetting scores across patients, surviving correction across the 13 structural/functional abnormalities examined (rho = 0.56, p-corr = 0.008); the mediation analysis demonstrated that this relationship held over and above the correlation of the mean GM volume of the right HPC cluster with the mean PCC rsALFF; there was thus a direct effect of reduced HPC GM volume on visual forgetting (within parenthesis: 95% confidence intervals); D and P: Doors and People (*Baddeley et al., 1994*); GM: gray matter; HPC: hippocampus; MAP: Memory and Amnesia Project; OPTIMA: Oxford Project To Investigate Memory and Aging; p: significance values are presented at uncorrected levels; PCC: posterior cingulate cortex; R: right (hemisphere); rsALFF: resting-state amplitude of low frequency fluctuations; TIV: total intracranial volume; VBM: voxel-based morphometry; z: age-scaled and standardized scores on Visual Forgetting (D and P); z-res: GM volumes from VBM clusters are residualized against age, sex, scan source (MAP, OPTIMA), and TIV across participants; mean rsALFF values are residualized across participants against age and sex.

DOI: https://doi.org/10.7554/eLife.46156.017

The following source data is available for figure 7:

**Source data 1.** This spreadsheet contains the mean.
DOI: https://doi.org/10.7554/eLife.46156.018

forgetting (rest of ps, p-corr ≥0.171). Nevertheless, at uncorrected levels, patients' reduced rsALFF in the PCC correlated with their visual forgetting scores (rho = 0.399, p-unc = 0.024), as well as with right HPC volume (r = 0.517, p=0.001). Right HPC volume correlated with visual forgetting scores over and above rsALFF in the PCC (partial correlation: rho = 0.460, p=0.009). The strong relationship of visual forgetting scores with right HPC volumes across patients was further demonstrated by a mediation analysis, whereby the effect of right HPC volume reduction on visual forgetting scores remained unmediated by the correlative reduction in PCC rsALFF across patients (direct effect: β = 0.43, p=0.036; indirect effect: β = 0.10, 95% CI: −0.11,0.32; *Figure 7*; *Figure 7—source data 1*).

Of the three composite memory scores, anterograde retention correlated with scores for depression (HADS) across patients (rho = −0.425, p-corr = 0.045; rest of ps, p-corr >0.248). We thus also examined the relationship among scores for depression, anterograde retention and HPC atrophy, with which anterograde retention scores strongly correlated. No correlation of scores for depression with HPC volumes (VBM clusters, manually delineated volumes) reached significance even at

uncorrected levels (all rhos, |rho| ≤ 0.25; all ps, p-unc ≥0.158), and right HPC volumes correlated with visual forgetting scores over and above depression scores (right HPC VBM cluster: rho = 0.564, p=0.001; manually delineated right HPC: rho = 0.530, p=0.002).

### Remote autobiographical memory

Remote autobiographical memory scores correlated across patients with their reduced left thalamic volume (r = 0.558, p-corr = 0.015), and only marginally with the volume expressed by the left HPC VBM cluster (r = 0.467, p-corr = 0.096). The volumes of the left thalamus and the left HPC VBM cluster correlated across patients (r = 0.47, p=0.003). Patients' reduced left thalamic volume correlated with their remote autobiographical memory scores over and above their reduced left HPC volume (VBM cluster) (partial correlation: r = 0.422, p=0.020). Moreover, the effects of left HPC atrophy on remote autobiographical memory were fully mediated by volume reduction in the left thalamus (direct effect: β = 0.94, p=0.165; indirect effect: β = 0.82, 95% CI: 0.20, 1.89; *Figure 8*; *Figure 8— source data 1*).

*Table 5* summarizes the relationships identified between memory impairment and structural and functional abnormalities in patients, as related to the effects of HPC atrophy. We did not identify any additional relationships when examining the volumes of manually delineated HPC portions (left/ right anterior/posterior HPC; Supplementary Table 5 in *Supplementary file 1*). Moreover, these relationships were not driven by the subset of patients (n = 7) who had not received immunosuppressive therapy (Supplementary Tables 7-8 in *Supplementary file 1*).

## Discussion

This study provides one of the largest investigations to date into the neural basis of 'hippocampal amnesia'. We examined the brain abnormalities underlying episodic memory impairment in autoimmune LE, a clinical syndrome in which acute, focal inflammation in the HPC leads, in the long-term, to HPC atrophy and amnesia. We hypothesized that HPC damage would be accompanied by remote effects on the structure and function of the extended HPC system (*Aggleton et al., 2010*), and that these abnormalities would be instrumental in explaining patients' anterograde and retrograde amnesia.

Consistent with previous investigations in smaller cohorts (*Butler et al., 2014*; *Finke et al., 2017*; *Henson et al., 2016*; *Irani et al., 2013*; *Malter et al., 2014*), our patients showed selective deficits in episodic memory, in the face of normal visuomotor, language, executive function, general and personal semantic memory performance. Episodic memory impairment was evident across a broad range of tests that have been extensively used in studies of MTL amnesia (*Bayley et al., 2003*; *Vann et al., 2009*). In terms of anterograde memory, patients showed impaired visual and verbal recall and recognition memory, and pronounced forgetting of visual material. Examining retrograde amnesia, we found striking loss of remote autobiographical memories from childhood and early adulthood.

The focal HPC high T2 signal on clinical MRI from the acute disease phase was followed by pronounced HPC atrophy and less pronounced right entorhinal cortical volume reduction within the MTL. However, patients also showed volume reduction in specific midline and lateral thalamic regions, along with reduced rsALFF in the PCC and precuneus, and reduced rsFC between the right HPC and the left HPC, the medial prefrontal and posteromedial cortices - regions known to interact closely with the HPC in the healthy brain.

These wider network abnormalities were associated with memory performance over and above HPC atrophy, and also fully mediated the relationships between HPC atrophy and memory performance. The only direct effect of HPC atrophy we observed was on forgetting, and no other brain abnormalities showed such an effect. Our results highlight the need to take into account remote changes in brain structure and function associated with HPC damage (*Aggleton, 2014*), since these may explain specific aspects of anterograde and retrograde amnesia and help identify others that may be a direct function of HPC atrophy per se.

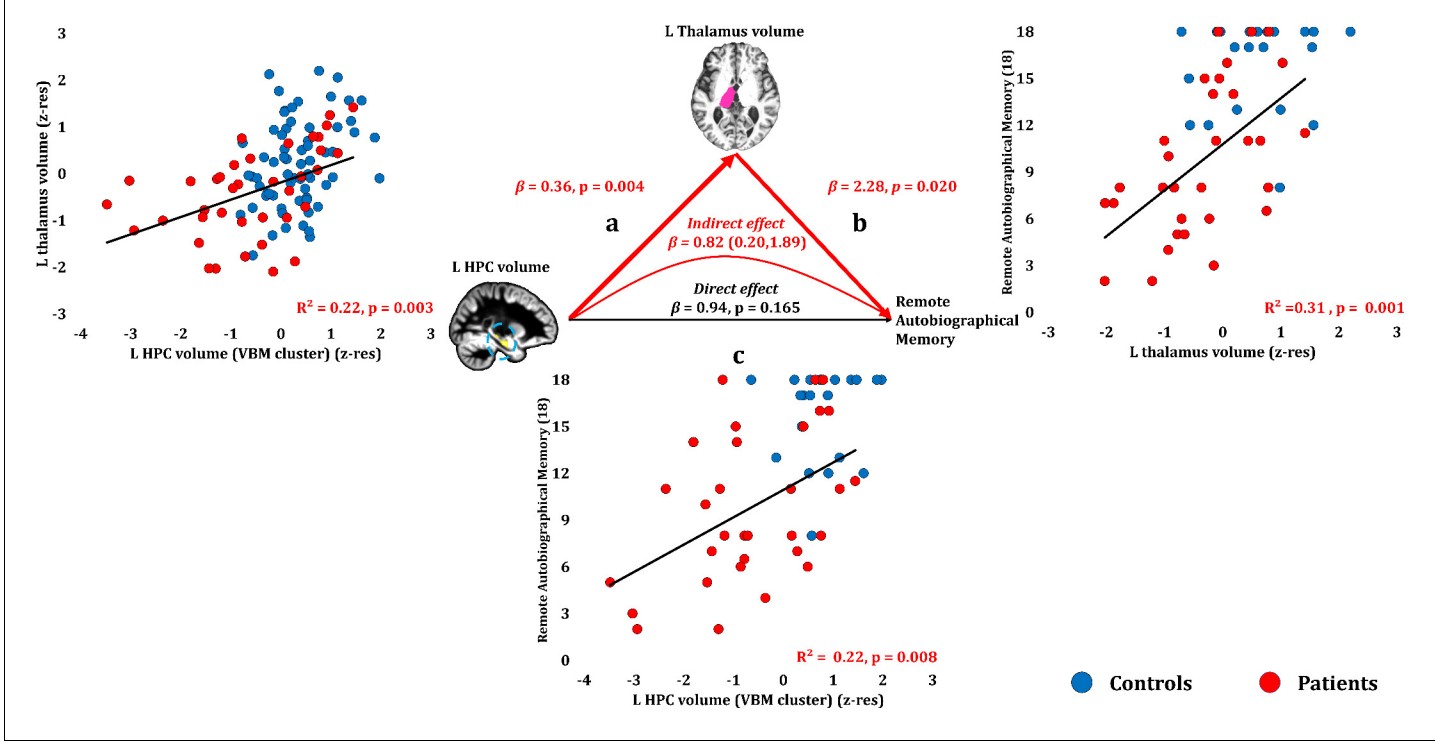

**Figure 8.** Remote Autobiographical Memory: Structural/Functional correlates. (**a**) GM volume of the left HPC VBM cluster correlated with the left thalamic volume across patients; (**b**) remote autobiographical memory (AMI) scores correlated across patients with the volume of the left thalamus, surviving correction across the 13 structural/functional abnormalities examined ($r = 0.558$, p-corr = 0.015); the mediation analysis demonstrates that this effect held when the correlation of thalamic volume with the volume expressed by the left HPC VBM cluster was accounted for; (**c**) the volume expressed by the left HPC VBM cluster correlated with remote autobiographical memory scores, albeit this correlation did not survive correction for multiple testing ($r = 0.467$, p-corr = 0.096); the mediation analysis demonstrated that this relationship did not hold over and above the correlation of the left thalamic volume with the HPC clusters; there was thus no direct effect of reduced HPC GM volume on remote autobiographical memory (within parenthesis: 95% confidence intervals); '18': remote autobiographical memory scores are the sums of the AMI scores for autobiographical memories for childhood and early adulthood (max = 18); AMI: Autobiographical Memory Interview; GM: gray matter; HPC: hippocampus; MAP: Memory and Amnesia Project; OPTIMA: Oxford Project To Investigate Memory and Aging; p: significance values are presented at uncorrected levels; TIV: total intracranial volume; VBM: voxel-based morphometry; z-res: volumes are residualized against age, sex, scan source (MAP, OPTIMA) and TIV across participants.

DOI: https://doi.org/10.7554/eLife.46156.019

The following source data is available for figure 8:

**Source data 1.** This spreadsheet contains the mean GM volume of the left HPC VBM cluster and the volume of the automatically delineated left thalamus (z-res) and the remote autobiographical memory scores [max = 18; Autobiographical Memory Interview; (*Kopelman et al., 1989*)] of healthy controls and patients (over the age of 50 at the time of assessment; See Materials and methods section) that are plotted in *Figure 8*; GM: gray matter; HPC: hippocampus; MAP: Memory and Amnesia Project; OPTIMA: Oxford Project To Investigate Memory and Aging; VBM: voxel-based morphometry; z-res: volumes are residualized against age, sex, scan source (MAP, OPTIMA), and TIV across participants. These data can be opened with Microsoft Excel or with open-source alternatives such as OpenOffice.

DOI: https://doi.org/10.7554/eLife.46156.020

## Abnormalities in the extended HPC system following HPC atrophy

We found bilaterally marked HPC atrophy and less significant right entorhinal cortical volume reduction using the gold standard of manual volumetry for MTL structures (*Grimm et al., 2015*). Automated segmentation of other subcortical structures disclosed left thalamic volume reduction.

However, our uniquely large cohort also enabled us to examine subtle structural abnormalities across the whole brain in a voxel-wise fashion. Using this method, volume reduction was observed in anterior-dorsomedial and dorsolateral thalamic regions, the nuclei within which are known to interact with the HPC and cingulate cortices (*Aggleton et al., 2010*). Our cross-sectional design cannot conclusively show that these extra-HPC abnormalities occurred as a consequence of HPC damage, an

**Table 5.** Summary of relationships between memory impairment and structural/functional abnormalities across patients.

| Memory composite score | | Effects of HPC atrophy |
|---|---|---|
| Anterograde retrieval | Verbal recognition | Does not explain additional variance beyond that explained by inter-HPC rsFC reduction |
| | Visual Recognition | Fully mediated by PCC rsALFF reduction |
| | Verbal Recall | |
| | Visual Recall | |
| Anterograde Retention (Visual Forgetting) | | Direct effect - not mediated by extra-HPC abnormalities |
| Remote Autobiographical Memory | | Fully mediated by thalamic volume reduction |

HPC: hippocampus; PCC: posterior cingulate cortex; rsALFF: resting-state amplitude of low frequency fluctuations; rsFC: resting-state functional connectivity.

DOI: https://doi.org/10.7554/eLife.46156.021

issue that needs to be addressed by future longitudinal studies. Nevertheless, acute inflammatory changes appeared confined to the HPC in the vast majority of our patients, while no such changes were found in the thalamus. Moreover, clinical T2-weighted MRIs in other autoimmune LE cohorts (*Finke et al., 2017*; *Kotsenas et al., 2014*; *Malter et al., 2014*), and the results of post-mortem studies have also demonstrated focal, acute HPC pathology in autoimmune LE patients (*Khan et al., 2009*; *Park et al., 2007*) and animal models (*Tröscher et al., 2017*). Our interpretation of these data is, therefore, that extra-HPC structures within the extended HPC system (thalamus, PCC) are affected as a secondary consequence of HPC damage, rather than due to the primary pathology. We also note that extra-HPC damage is equally if not more likely in other conditions associated with HPC amnesia, such as cases of ischemia/anoxia (*Huang and Castillo, 2008*). It should, however, be acknowledged that a single case study of amnesia due to ischemia/anoxia which came to post mortem demonstrated focal neuronal loss in the CA1 region of the HPC on histopathology (*Zola-Morgan et al., 1986*). Direct comparisons between autoimmune LE and amnesia due to different etiologies would be useful in future research.

Patients also showed functional abnormalities in the posteromedial cortex, and the relationship between HPC atrophy and PCC rsALFF was fully mediated by the correlative volume reduction in the thalamic regions. This is consistent with the functional abnormalities previously observed in posteromedial cortical regions in both HPC and diencephalic amnesias in humans (*Aggleton, 2014*; *Reed et al., 1999*). Patients also showed reduced inter-HPC rsFC. In healthy individuals, the spontaneous activity of the HPC is coupled with that of the contralateral HPC at rest and these regions form part of the default-mode network. This is also the case for the medial prefrontal and posteromedial cortical regions which showed reduced rsFC with the HPC (*Buckner et al., 2008*; *Greicius et al., 2004*). The decrease in inter-HPC rsFC across patients as a function of delay since symptom onset is consistent with the idea that these network-wide abnormalities follow focal HPC damage. In future, this needs replication in longitudinal studies.

## Anterograde memory: Retrieval

Patients showed impairment across all tests of visual and verbal recall and recognition memory. The only unimpaired form of memory was face recognition. This would be predicted by some material-specific accounts of recognition memory (*Bird and Burgess, 2008*) and is consistent with the view that recognition of unfamiliar faces is exceptional in several regards – according to some approaches, faces may be holistically processed and difficult to label verbally, and adequate performance levels may be attained in the absence of capacities to associate faces with a study list (*Smith et al., 2014*). Ours is the largest study to date to confirm preserved face recognition memory in HPC amnesia.

At uncorrected levels, HPC volumes correlated with both recall and recognition memory scores. Crucially, however, the reduction in PCC rsALFF correlated with verbal and visual recall and visual recognition over and above HPC atrophy and, moreover, fully mediated the effects of HPC atrophy on these aspects of episodic memory. This finding demonstrates that the HPC role in recall and

recognition cannot be examined in isolation from remote effects of HPC damage within the extended HPC system. It furthermore dovetails with the causal role that has been attributed to the posteromedial cortex in episodic memory for both visual (*Bonnì et al., 2015*) and verbal (*Koch et al., 2018*) material, and aligns with evidence from neurodegenerative diseases that it is posteromedial cortex rather than HPC function that predicts episodic memory impairment (*La Joie et al., 2014*). Patients' impaired verbal recognition was associated with reduced inter-HPC rsFC. The reasons why the correlates of verbal recognition were different from those of other forms of memory are unclear, but the relationship with inter-HPC rsFC is consistent with that found between inter-HPC connectivity and episodic memory in healthy (*Wang et al., 2010*) and patient populations (e.g. traumatic axonal injury) (*Marquez de la Plata et al., 2011*).

It should be noted that our neuropsychological assessment did not explicitly attempt to distinguish between the recollection and familiarity processes that may underlie differences in recall and recognition memory. In order to identify the neural mechanisms supporting these processes, finer-grained behavioral paradigms are required. Furthermore, our study did not include HPC subfield volumetry, which may have disclosed stronger correlations between HPC subfields and memory scores. The volumes and hemodynamic activity of HPC subfields have been associated with various forms of memory (*Bonnici et al., 2013*; *Miller et al., 2017*; *Palombo et al., 2018*). While imaging at higher field strength is required to investigate HPC subfields, our study highlights the fact that the remote effects of HPC damage on wider brain networks are a key factor in HPC amnesia.

It is important to acknowledge an alternative interpretation of the relationship that we observed between resting-state functional abnormalities and memory measures. McCormick and colleagues recently showed that HPC amnesic patients differ in the form and content of their 'mind-wandering' compared with healthy adults (*McCormick et al., 2018*). It is possible that 'resting-state' functional abnormalities in our patients actually reflect (and are perhaps therefore mediated by) differences between healthy controls and patients with respect to the extent of mind wandering in the scanner, rather than differences in neurovascular functioning in these regions per se. Anterior and posterior midline structures are strongly implicated in mind-wandering. Nevertheless, rsfMRI measures (predominantly rsFC) have repeatedly provided reliable correlates of memory impairment in HPC amnesia [e.g. (*Heine et al., 2018*; *Henson et al., 2016*)]. Moreover, this interpretation is not inconsistent with the basic premise of our argument, namely, that the effects of HPC damage on memory are mediated by other processes that are compromised following HPC damage. Our study was not designed to disambiguate the level of disruption that is mediating the effects of HPC atrophy. In other words, the disruption could be at the cognitive level, or at the neurovascular level, or at both levels. Further work is needed to disambiguate those two interpretations.

## Anterograde memory: Forgetting

The only aspect of amnesia upon which HPC atrophy showed a direct effect was that of visual forgetting (of abstract shapes). Patients showed only a marginal increase in verbal forgetting. In particular, it was the atrophy in the right HPC that was associated with visual forgetting. This is consistent with evidence for HPC lateralization of maintenance processes for verbal (left HPC) (*Frisk and Milner, 1990*) and visual material (right HPC) (*Smith and Milner, 1989*). Indeed, rapid forgetting of visual information in right HPC damage was noted quite early (*Jones-Gotman, 1986*), as was, in general, the nature of forgetting in HPC amnesia (*Huppert and Piercy, 1979*). This has recently been re-emphasized (*Sadeh et al., 2014*), with functional neuroimaging studies in healthy young adults attributing a central role to the HPC in constraining the forgetting that occurs by new learning (*Kuhl et al., 2010*).

Our results go beyond these studies and demonstrate an exclusive relationship between HPC atrophy and rapid forgetting even within the context of abnormalities in the extended HPC system. This relationship is consistent with computational models of HPC function that outline pattern completion and pattern separation (*O'Reilly and McClelland, 1994*) as two core mechanisms that protect against forgetting. Pattern completion enables the reinstatement of previously encoded memories from incomplete input, whereas pattern separation involves the orthogonal coding of memories for overlapping events, which minimizes forgetting by generating non-interfering representations (*Kuhl et al., 2010*).

These findings suggest that the HPC role in amnesia may be much less direct than has previously been held, at least with respect to performance on widely used, standardized neuropsychological

tests of memory. In contrast with the majority of such tests, the measurements of forgetting such as that used here partially control for factors that may confound simple recall scores, such as variability in attention, initial learning or retrieval strategy, factors that may depend on regions outside the HPC. Instead, measurements of forgetting may provide the most sensitive and specific way to detect HPC damage. This proposal does not preclude the possibility that the HPC mechanisms underlying the retention of newly formed memories are process- or material-specific. As our forgetting rates were derived only from a recall-based test (*Baddeley et al., 1994*), we are unable to comment on the relationship of HPC atrophy with forgetting in recognition memory, a relationship which has recently been questioned (*Sadeh et al., 2014*). Likewise, material-specificity in rapid forgetting should be examined with finer-grained behavioral tasks, involving different types of stimuli (e.g. scenes vs. faces). Nevertheless, the direct relationship of (right) HPC atrophy with (visual) forgetting highlights the utility of tests of accelerated forgetting in detecting HPC pathology in other disorders. Recent studies have emphasized a possible role for such tests in the early detection of Alzheimer's disease pathology (*Weston et al., 2018*; *Zimmermann and Butler, 2018*).

## Remote autobiographical memory

Patients showed spared remote personal semantic memory, in the face of impaired remote autobiographical memories. To date, research in autoimmune LE patients has focused on anterograde memory (*Butler et al., 2014*; *Finke et al., 2017*; *Henson et al., 2016*; *Malter et al., 2014*; *Miller et al., 2017*), with very little evidence on retrograde amnesia (*Chan et al., 2007*). Our study is also one of the very few to examine the relationship of HPC damage with both anterograde and retrograde amnesia.

It is possible that the test we used to assess remote memories (AMI) is insufficiently sensitive to truly 'episodic' aspects of autobiographical memory that have been held to be HPC-dependent (*Moscovitch et al., 2016*; *Nadel and Moscovitch, 1997*). Other methods, involving parametric text-based procedures, may offer greater sensitivity [e.g. the Autobiographical Interview (*Levine et al., 2002*)]. Nevertheless, studies on other amnesic cohorts have found the AMI and the Autobiographical Interview to offer comparable results (*Rensen et al., 2017*).

Our findings would at first sight appear to support a role for the HPC in the recollection of remote autobiographical memories. However, the volume of the left thalamus correlated with remote autobiographical memory scores over and above HPC volumes across patients. Moreover, the effects of HPC atrophy were fully mediated by the volume reduction in the thalamus. Furthermore, no selective relationship was found between remote autobiographical memory scores and anterior or posterior HPC volumes. These findings are difficult to reconcile with the multiple trace (*Moscovitch et al., 2005*) or trace-transformation accounts of retrograde amnesia (*Sekeres et al., 2018*), according to which remote autobiographical memory deficits should be a function of the extent of HPC damage, either *in toto*, or in specific HPC portions.

The absence of direct effects of HPC atrophy on remote memory, in combination with the selective presence of such effects on forgetting rates, is instead consistent with the idea that HPC circuitry acts as a buffer for the temporary maintenance of episodic information, with non-HPC structures storing more permanent memory traces. Our findings may thus offer some support for the standard consolidation model of remote memory. Other work supporting this model has primarily emphasized the role of neocortical damage in retrograde amnesia (*Bayley et al., 2003*; *Squire et al., 2015*). We here highlight the need also to consider the role of the thalamus. Focal thalamic damage has been often associated with retrograde amnesia in single-case or small case-series studies (*Carlesimo et al., 2011*; *Stuss et al., 1988*), and mediodorsal thalamic nuclei may coordinate the recollection of cortically stored memories (*Miller et al., 2001*) via thalamic-prefrontal cortical projections (*Kopelman, 2015*). Our results show that secondary alterations in thalamic integrity may explain the large variability in retrograde memory loss reported after HPC damage (*Bartsch et al., 2011*; *Bayley et al., 2003*; *St-Laurent et al., 2011*).

Finally, our finding that thalamic volume was reduced in our patient group is consistent with the literature on developmental amnesia due to early hypoxic-ischaemic encephalopathy, where HPC damage has been noted along with atrophy in the thalamus and the mammillary bodies [e.g. (*Dzieciol et al., 2017*)]. A question for future research is therefore whether adult-onset HPC damage is accompanied by atrophy in the mammillary bodies, or whether thalamic atrophy may occur in the absence of changes to the mammillary bodies, given the evidence for direct forniceal projections

from the HPC/subiculum to the anterior thalamus in human and non-human primates (*Bubb et al., 2017*).

## Conclusion

Our study, probably the largest in human HPC amnesia, shows that abnormalities in the integrity of and connectivity within the extended HPC system may occur after focal HPC damage, and that these play a central role in explaining the variability in the anterograde and retrograde amnesia of patients with HPC atrophy. We found that the neuropsychological measure most sensitive and specific to HPC atrophy was the forgetting of newly learned information. Understanding the impact of network-wide changes following HPC damage will be key to resolving long-standing controversies in the literature on human amnesia.

## Materials and methods

### Participants

#### Patients

##### Inclusion and exclusion criteria

We identified 38 patients (26M:12F; age at research MRI: M = 63.06; IQR = 16.16 years; *Table 2*) who i) had undergone neuropsychological assessment during the acute phase of the illness at the Russell Cairns Unit, Oxford (2013–2018); ii) had been diagnosed with LE according to the diagnostic criteria described in (*Graus et al., 2016*): a) subacute onset of memory deficits [(*Graus et al., 2016*) mention 'working memory deficits'. However, other studies have used 'short-term memory' as a criterion (*Graus, 2004*) and the classical deficit is generally held to be in episodic memory (*Vincent et al., 2004*)], seizures, or psychiatric symptoms suggesting involvement of the limbic system; b) bilateral brain abnormalities on T2-weighted MRI, restricted within the MTL; c) CSF pleocytosis (white blood cells > $5/mm^3$) and/or EEG with epileptic or slow-wave activity involving the temporal lobes; d) reasonable exclusion of alternative causes (e.g. CNS infections, septic encephalopathy, metabolic encephalopathy, drug toxicity, cerebrovascular disease, neoplastic disorders, Creutzfeldt-Jakob disease, epileptic disorders, rheumatologic disorders, Kleine-Levin, mitochondrial diseases); e) detection of antibodies against cell-surface, synaptic, or onconeural proteins. According to *Graus et al. (2016)*, criteria (a-d) are required for a diagnosis of 'definite LE', unless, in the absence of one of (a-c), criterion (e) is satisfied.

34/38 patients met the criteria for a diagnosis of 'definite LE'. The remaining four patients had been diagnosed with autoimmune LE, and satisfied criteria (a), (b), and (d), but not (e), and no data could be recovered on (c). In the majority of patients (28/38), an autoantibody known to be associated with LE had been identified, but 10/38 had the typical clinical profile of LE with no identified antibody. Such cases are well-recognised (*Graus et al., 2018*) and are likely here due to antibodies not detectable in routine clinical practice at the time of screening. No patient presented with NMDAR-antibody encephalitis (*Malter et al., 2013*) or positive polymerase chain reaction testing for herpes simplex virus. Only two cases (28,31) had identifiable neoplastic lesions that were treated and were in full remission at the time of study participation. 31/38 patients had been treated with immunotherapy (e.g. intravenous and/or oral prednisolone, plasma exchange) in the acute phase of the illness.

Furthermore, all patients i) had undergone MRI at the time of initial clinical presentation; ii) were fluent in English (37 native speakers; one non-native speaker); iii) had no history of previous psychiatric or neurological disorder that could have resulted in cognitive impairment and iv) had no contraindication to MRI at the time of entry into the study.

Patients were all recruited in the post-acute phase of the illness (M = 5.41; IQR = 5.36 years since symptom onset) and were re-assessed by an experienced neurologist (CRB) prior to study inclusion.

#### Acute clinical MRI

Neuroradiological reports of patients' MRI scans from the time of initial clinical presentation were consulted to identify abnormalities in T2 signal, volume, and diffusion, within and beyond the MTL. 34/38 patients showed abnormal signal, volume (swelling or atrophy) and/or (facilitated) diffusion in the HPC. High signal in the amygdala was noted in 6/38 patients. One patient showed abnormalities

in the entorhinal cortex, and one patient in the parahippocampal cortex (*Table 2*). No acute abnormalities were detected in 4/38 patients who nonetheless had clinical features characteristic of autoimmune LE. Only one patient showed extra-MTL abnormalities (bright caudate), and four patients had mild microangiopathic changes, which are commonly found with aging.

## Healthy controls

Healthy controls were recruited through local advertisement and through the Oxford Project To Investigate Memory and Aging. Of the healthy controls that underwent neuropsychological assessment (see below), all were fluent in English (40 Native speakers; one non-native speaker; Supplementary Table 9 in *Supplementary file 1*).

All participants provided written informed consent according to the Declaration of Helsinki. Ethical approval was received from South Central Oxford Research Ethics Committee (REC no: 08/H0606/133).

## Neuropsychological assessment

All 38 patients (26M:12F; age at assessment: M = 61.32; IQR = 15.82 years) were assessed with neuropsychological tests, along with 41 control participants (27M:14F; age at assessment: M = 61.50; IQR = 16.93 years; controls vs. patients: age: U = 706, p=0.477; M:F ratio: $\chi^2$ = 0.059, p=0.808).

Neuropsychological tests were administered to assess the following domains: premorbid intelligence [National Adult Reading Test (NART) (*Nelson and Willison, 1991*)]; executive function [Wechsler Memory Scale III (WMS-III) digit span (*Wechsler, 1997*); Delis-Kaplan Executive Function System (DKEFS) Trails (*Delis et al., 2001*)]; semantic memory and language [Camel and Cactus Test (C and CT) (*Bozeat et al., 2000*); Graded Naming Test (GNT) (*McKenna and Warrington, 1980*); Weschler Abbreviate Scale of Intelligence (WASI/WASI-II) Vocabulary and Similarity (*Wechsler, 2011*)]; visuospatial and motor function [Rey-Osterrieth Complex Figure Test (ROCFT) copy (*Rey, 1959*); DKEFS trails visual scanning, motor speed (*Delis et al., 2001*); Visual Object and Space Perception (VOSP) cube analysis, dot counting, position discrimination (*Warrington and James, 1991*)]; anxiety and depression [Hospital Anxiety and Depression Scale (HADS) (*Zigmond and Snaith, 1983*)].

### Anterograde memory

Anterograde memory was examined using a range of memory tests: WMS-III (*Wechsler, 1997*); ROCFT (*Rey, 1959*); the Warrington Recognition Memory Tests for faces and words (*Warrington, 1984*) and the Warrington Topographical Memory test for scenes (RMT) (*Warrington, 1996*); the Doors and People test (D and P) (*Baddeley et al., 1994*).

### Retrograde memory

In order to assess patients' remote autobiographical memory, we administered the most commonly used clinical test of autobiographical memory, the Autobiographical Memory Interview [AMI; (*Kopelman et al., 1989*)]. The AMI assesses semantic autobiographical memory ('personal semantics') through a structured interview concerning autobiographical facts. Episodic autobiographical memory is examined through the free recall of events and is scored (0–3) for specificity and richness of episodic recall. Following the AMI manual, three points were given for episodic memories produced in a highly detailed fashion, with mention of specific time and place; 0–2 points were given to memories recalled with poor episodic detail, that is two points were credited when the episodic memory was not recalled in detail or did not involve mention of time and place, or where the memory was not recalled; one point was given for a vague personal memory; and 0 points when a response was not provided or when general semantic information was provided instead (*Kopelman et al., 1989*). Memories of three periods are assessed: childhood (up to 18 years of age), early adulthood (up to participants' mid 30's), and recent memories (0–5 years from the time of the interview). We identified 31 patients (23M:8F; age at assessment: M = 63.79, IQR = 15.16 years) and 29 controls who were over the age of 50 at the time of assessment (19M:10F; age: M = 65.74, IQR = 17.23; controls vs. patients: age: U = 431.00, p=0.789; M:F ratio: $\chi^2$ = 0.537, p=0.464). As such, recent memories did not overlap with those of early adulthood (recent autobiographical/semantic memories probed by this test pertain to 0–5 years before the time of assessment), and

involved a minimum of 10 years between the earliest recent memory and the latest memory for early adulthood that the AMI probed.

Scores in the 'recent memory' conditions, which cover a period of 0–5 years preceding the date of assessment, at least partly reflected anterograde memory impairment, as they overlapped with the period post-disease onset (time since acute presentation (mean = 5.99; SD = 4.00 years). Indeed, of the 31 patients analyzed, only one reported recent memories formed premorbidly. The recent memory scores of the AMI could thus not capture retrograde amnesia for recent personal semantic and autobiographical memories, and were not considered in the analysis.

For the purposes of data reduction, minimization of measurement error, maximization of generalizability of our findings, and simplification of correlational analyses, we used i) an anterograde memory composite score for visual recall (D and P Shapes, ROCFT Immediate and Delayed Recall), verbal recall (WMS-III Logical Memory I,II, Word List I,II, D and P People), visual recognition (RMT Scenes, D and P Doors), and verbal recognition memory (RMT Words, D and P Names, WMS-III Word List Recognition), reflecting 'anterograde retrieval', by averaging the age-scaled, standardized scores of each patient across the tests of anterograde memory in which patients showed group-level impairment; ii) a remote autobiographical memory score, by summing patients' AMI scores for autobiographical memories in childhood and early adulthood; iii) the D and P scores for verbal and/or visual forgetting - unlike other tests of immediate and delayed recall memory that involve no mechanism for equating participants on immediate recall, the D and P quantifies forgetting on the basis of participants' performance in the last trial of immediate recall relative to their performance in delayed recall.

## Brain imaging
### Scanning procedures
Image acquisition was conducted on a Siemens 3T Trio system using a 32-channel head coil (University of Oxford Centre for Clinical Magnetic Resonance Research).

### Structural MRI
3D T1-weighted images were acquired using a Magnetization Prepared Rapid Gradient Echo (MPRAGE) sequence (echo time = 4.7 ms, repetition time = 2040 ms, 8° flip angle, field of view = 192 mm, voxel size = $1\times1\times1$ mm). All 38 patients (26M:12F; age at imaging: M = 63.06; IQR = 16.06 years) underwent structural brain imaging, along with 67 control participants (35 recruited by the Memory and Amnesia Project; 32 datasets were made available through the Oxford Project To Investigate Memory and Aging; 40M:27F; age at imaging: M = 64.70; IQR = 19.87 years; controls vs. patients: M:F ratio: $\chi^2$ = 0.79, p=0.374; age at imaging: U = 1239.5, p=0.825).

### Resting-state fMRI
BOLD-weighted fMRI data were acquired using a Gradient Echo EPI sequence. Participants were instructed to lay still, keep their eyes open to watch the fixation cross presented on the in-scanner projector, and not to fall asleep. Whole-brain data were acquired (180 volumes; slice thickness = 3.5 mm, echo time = 28 ms, repetition time = 2410 ms, 89° flip angle, field of view = 192 mm, voxel size = $3\times3\times3.5$ mm). 35/38 patients (three datasets were discarded due to acquisition errors and/or movement; 24M:11F; age at imaging: M = 61.45; IQR = 15.85 years) underwent resting-state fMRI, along with 32 control participants (three datasets discarded due to acquisition errors and/or movement; only structural MRIs were available for the remaining 32 controls that were made available through the Oxford Project To Investigate Memory and Aging; 23M:9F; median = 55.71; IQR = 17.18 years; controls vs. patients: M:F ratio: $\chi^2$ = 0.087, p=0.768; age at imaging: U = 425.00, p=0.091).

## GM volume, hemodynamic activity, and functional connectivity
### Volumetry
#### Manual segmentation of MTL
Manual segmentation of the MTL (38/38 patients; 48/67 healthy controls) was carried out in native space using ITK-SNAP (*Yushkevich et al., 2006*) by a trained researcher (ARF) (protocol: https://www.ndcn.ox.ac.uk/files/research/segmentation_protocol_medial_temporal_lobes.pdf).

Segmentation procedures were based on published atlases and protocols (*Insausti et al., 1998*; *Pruessner et al., 2002*). The structures delineated were the HPC, amygdala, and the perirhinal, entorhinal, parahippocampal, and temporopolar cortices. Structures were segmented in each hemisphere. As originally reported in *Loane et al. (2019)*, intra-rater reliability was measured using intra-class coefficient correlations [ICC(3)] on a random selection of 24 hemispheres (12 patients and 12 controls) segmented twice by ARF with an interval of at least 4 months between segmentations. Intra-rater reliability matched that reported for manual MTL segmentations (*Olsen et al., 2013*) (HPC: 0.98; amygdala: 0.80; perirhinal cortex: 0.90; entorhinal cortex: 0.82; parahippocampal cortex: 0.93; temporopolar cortex: 0.96).

### Automated segmentation of other subcortical structures

The brainstem, thalamus, caudate nucleus, putamen, nucleus accumbens and pallidum were also automatically segmented using FSL-FIRST (v. 6.0; http://www.fmrib.ox.ac.uk/fsl) (*Patenaude et al., 2011*) for all participants.

We compared healthy controls and patients on the volumes of all structures in a series of ANCO-VAs, using age, sex, scan source [MAP (Memory and Amnesia Project), or OPTIMA (Oxford Project To Investigate Memory and Aging)], and total intra-cranial volume (TIV; derived from the unified segmentation procedure in SPM12; see below) as between-subjects covariates. Volumes that were reduced in our patient group were residualized against these four variables (z-res) and entered in a series of bivariate correlations with patients' memory scores.

## Voxel-based morphometry

To identify GM volume discrepancies between groups at a whole-brain level, the T1-weighted images were analysed with VBM (*Ashburner and Friston, 2000*), using the Statistical Parametric Mapping software (SPM12 v7219; http://www.fil.ion.ucl.ac.uk/spm/software/spm12) running in Matlab R2017b.

Images were examined for scanner artefacts, reoriented to have the same point of origin (anterior commissure) and spatial orientation, bias-corrected to remove intensity non-uniformities, and segmented into GM, white matter (WM), and cerebrospinal fluid (CSF), using the unified segmentation procedure (*Ashburner and Friston, 2005*). The diffeomorphic anatomical registration through the exponentiated lie algebra (DARTEL) toolbox was applied to all participants' GM, WM, and CSF to refine inter-subject registration, and study-specific GM templates were generated (*Ashburner, 2007*). After affine registration of the GM DARTEL templates to the tissue probability maps in MNI (Montreal Neurological Institute, Quebec, Canada) space, non-linear warping of GM images was performed to the DARTEL GM template in MNI space. Voxel values in the tissue maps were modulated by the Jacobian determinant (calculated during spatial normalization), with modulated GM images reflecting tissue volume. These images (voxel size: 1 mm$^3$ isotropic) were smoothed using a standard Gaussian filter (4 mm FWHM) to increase spatial specificity within MTL structures. We compared GM volume between groups, including age, sex, TIV, and scan source (MAP, OPTIMA) as second-level covariates. We report clusters surviving FWE-correction ($p < 0.05$) at peak-voxel level over $p < 0.001$ (uncorrected).

The mean GM volumes of VBM clusters were extracted using the Marsbar toolbox in SPM12 (*Brett et al., 2002*), were residualized across participants against age, sex, scan source (MAP, OPTIMA), and TIV, and were used in a series of bivariate correlations with memory scores across patients.

## fMRI analysis

Resting-state fMRI preprocessing and connectivity analyses were conducted using the CONN toolbox version 18.a (https://www.nitrc.org/projects/conn) (*Whitfield-Gabrieli and Nieto-Castanon, 2012*) in SPM.

### Preprocessing

The EPIs were spatially realigned to correct for interscan movement and were slice time-corrected. The structural MRIs were coregistered to the EPIs, segmented and normalized along with EPIs in MNI space, followed by motion outlier detection (ART-based scrubbing). Denoising including the

anatomical component-based correction method (CompCor) (*Behzadi et al., 2007*) was used to remove sources of noise in the BOLD time series data, deriving principal components from WM and CSF. WM, CSF and the six movement parameters were included as first-level nuisance covariates. A temporal band pass filter (0.01–0.1 Hz) was applied to this residual BOLD signal to remove low-frequency drift and high-frequency respiratory and cardiac noise. Images were smoothed using a Gaussian filter (8 mm FWHM).

We explored resting-state functional abnormalities in patients with respect to both segregation and integration, that is in terms of hemodynamic activity in local regions and functional connectivity between regions. While the majority of rsfMRI studies investigate the correlations between brain areas from the perspective of integration, that is 'resting-state functional connectivity' (rsFC) (providing holistic information on sets of interacting brain regions) and the abnormal integration between brain areas in patient groups relative to healthy controls, these approaches do not directly provide information on the specific brain regions that show abnormalities in patients (in the form of the amplitude of spontaneous brain activity). The latter are reliably indexed by the resting-state amplitude of low-frequency fluctuation (rsALFF) of the rsfMRI signal, and help us identify the specific brain regions of abnormal spontaneous activity (*Zang et al., 2007*; *Zou et al., 2008*), similar to resting-state CBF and glucose metabolic rate in PET studies. For the above reasons, the combined application of these two methods (rsFC and rsALFF) has been held to provide more information than either method alone (*Lv et al., 2018*). We used data-driven approaches in both.

The same pre-processing pipeline was used for both rsALFF and rsFC analyses, including denoising with a temporal band pass filter (0.01–0.1 Hz), as in the majority of studies examining rsALFF [e.g. (*Cui et al., 2014*; *Dai et al., 2012*; *Satterthwaite et al., 2012*; *Yao et al., 2012*; *Yi et al., 2012*)].

## Resting-state hemodynamic activity: rsALFF

We examined local abnormalities in the intensity of slow spontaneous fluctuations of hemodynamic activity at rest across the whole brain, using an analysis of rsALFF (*Zang et al., 2007*). RsALFF is defined as the total power within the low-frequency range (0.01–0.1 Hz), and thus indexes the strength or intensity of low frequency oscillations; rsALFF has been linked to neuronal glucose metabolism (*Tomasi et al., 2013*) and correlates with local field potential activity (*Logothetis et al., 2001*). Alterations in rsALFF have been shown in a number of disorders (*Küblböck et al., 2014*; *Kwak et al., 2012*; *Lui et al., 2015*), and can also reflect individual differences in performance in a large variety of cognitive tasks (*Mennes et al., 2011*; *Wei et al., 2012*). We opted for ALFF over fALFF (fractional amplitude of low frequency fluctuations, that is the total power within the low-frequency range, divided by the total power in the entire detectable frequency range), as the former demonstrates higher test–retest reliability in GM regions, which suggests that it is potentially more sensitive for discerning differences between individuals and groups (*Zuo et al., 2010*).

## Resting-state functional connectivity: Multivariate pattern analysis

Instead of selecting seed/target ROIs or networks in an a priori fashion for our rsFC analyses, we chose to capitalize on the size of our patient cohort and use a PCA-based approach, commonly referred to as 'connectome-MVPA' ('multivariate pattern analysis') [e.g. (*Arnold Anteraper et al., 2019*; *Flodin et al., 2016*; *Kazumata et al., 2017*; *Thompson et al., 2016*; *Whitfield-Gabrieli et al., 2016*; *Yankouskaya et al., 2017*); implemented in the Conn toolbox (*Whitfield-Gabrieli and Nieto-Castanon, 2012*) https://sites.google.com/view/conn/measures/networks-voxel-level], in order to identify seed regions for post-hoc seed-to-voxel connectivity analyses in a data-driven fashion. This method has been extensively used to identify the regions in which groups of patients differ significantly from healthy controls with respect to their rsFC with the rest of the brain (e.g. *Flodin et al., 2016*; *Kazumata et al., 2017*; *Thompson et al., 2016*). As outlined in other papers [e.g. (*Whitfield-Gabrieli et al., 2016*)], the strength of this approach is the use of a massive data set (connectivity between all pairs of recorded voxels) for the purposes of identifying the most reliable difference in rsFC across the whole brain between groups of participants. In other words, it is an agnostic, unbiased approach used to define data-driven regions of interest (seeds) prior to performing a post-hoc analysis on the seeds to analyse brain connectivity patterns. This PCA-based method is more reproducible than conventional seed-based approaches (*Song et al., 2016*). As this

approach is both data-driven and conducted across the whole brain, it is strongly preferable to approaches examining a priori defined networks or seed/target regions of interest, commonly employed by studies of HPC amnesia with substantially smaller sample sizes.

In particular, the 'connectome-MVPA' method assesses the multivariate pattern of pairwise connections between voxels across the whole brain by performing a PCA separately for each voxel that characterizes its rsFC with the rest of the brain, in two steps. In the first step, separately for each participant, a default number (n = 64) of Singular Value Decomposition (SVD) components characterizing each participant's voxel-to-voxel correlation matrix is retained. The resulting component scores are stored as first-level voxel-to-voxel covariance matrices for each participant. In the second step, a low-dimensional representation of the entire pattern of seed-based correlations between this voxel and the rest of the brain is derived for each voxel across participants by retaining a certain number of principal components that explain most of the variance of the connectivity matrix. In our study, we retained the first seven principal components, in keeping with a conventionally used conservative 1:10 ratio between the number of components extracted and the number of subjects (n = 67). These seven resulting component score volumes best represented the whole-brain connectivity pattern for each participant, explaining the maximum inter-subject variability. They were simultaneously included in a second-level analysis F-test at group-level (an omnibus test, equivalent to seed-level F-tests in ROI-to-ROI analyses of rsFC), testing for clusters that differ between healthy controls and patients with respect to whole-brain connectivity, as represented by the PCA component volumes, while also including age and sex as between-subjects covariates. This method was then followed by post-hoc analyses to determine specific connectivity patterns in the data.

## Resting-state FC: Seed-to-voxel connectivity analysis

The omnibus F-test above was followed up by post-hoc analyses to determine specific connectivity patterns in the data. We therefore conducted a whole-brain seed-to-voxel rsFC analysis, seeding from the regions identified from the omnibus F-test above (controls vs. patients; covariates: age, sex), to explore connectivity between those regions and the rest of the brain.

Both seed-to-voxel rsFC and rsALFF analyses involved a t contrast (controls > patients; covariates: age, sex), with statistical parametrical connectivity maps thresholded at a voxel level of $p<0.001$ and FWE-corrected ($p<0.05$) at cluster- or voxel peak-level.

## Structure/Function-Behavior Correlations

Given our a priori hypothesis regarding the relationship of HPC atrophy with memory impairment, correlations of HPC volumes (GM volume expressed by HPC VBM clusters or manually delineated HPC volumes) with memory scores were investigated at uncorrected levels (p-unc <0.05). Correlations of extra-HPC abnormalities with memory scores were corrected for multiple testing (see below).

We examined the relationship of each abnormality with patients' anterograde retrieval (verbal and visual recognition, verbal and visual recall) composite scores, their overall remote autobiographical memory scores [AMI (*Kopelman et al., 1989*)], and their visual or verbal forgetting scores from the Doors and People test (*Baddeley et al., 1994*).

Given our hypothesis that HPC atrophy gives rise to a series of correlative abnormalities in regions within broader networks (e.g. the HPC-diencephalic-cingulate networks) that may themselves be associated with memory impairment, we conducted a series of partial correlation analyses between memory composite scores and extra-HPC structural/functional abnormalities, controlling for HPC volumes. We also conducted a series of mediation analyses (see below): HPC volumes were entered as the independent variables, and memory scores as the dependent variables. Mediator variables were structural or functional abnormalities that were correlated with both HPC volumes and memory scores across patients.

Measures of structural or functional abnormality that did not correlate with each other across patients but were both associated with impaired memory scores were entered as independent variables in a series of multiple step-wise linear regression analyses (dependent variable: memory scores), in order to investigate the portion of the variance of memory impairment that could be explained by each of those abnormalities (see below).

## Experimental design and statistical analysis

Non-imaging statistical analyses were performed using SPSS (version 25.0, SPSS Inc). Variance homogeneity was assessed using Levene's test, and normal distribution using the Shapiro-Wilk test. Parametric (Student t-test; Welch t-test used when the assumption of homogeneity of variances was violated) and non-parametric tests (Mann-Whitney U employed when the assumption of normal distribution was not met in a group) were used appropriately. Pearson correlation coefficient (*r*) and Spearman's rho ($\rho$) (when variables were not normally distributed) were used to examine the relationship among measures of structural, functional abnormality, and memory scores in a series of bivariate and partial correlations. Significance values were corrected ('p-corr') for multiple comparisons and correlations with the Holm-Bonferroni sequential correction method (*Holm, 1979*), unless otherwise stated ('p-unc'), as in the case of correlations between HPC volumes and memory scores. In cases where correction for multiple comparisons does not apply, 'p' was used instead.

## Mediation analyses

Mediation is a hypothesis about a causal relation among variables (*Judd and Kenny, 1981*; *Baron and Kenny, 1986*; *MacKinnon et al., 2007*). Four conditions are required to be met to establish mediation: i) the independent variable must be associated with the dependent variable; ii) the independent variable must be associated with the mediator; iii) the mediator variable must be associated with the dependent variable; iv) the mediator variable mediates the relationship between the independent and the dependent variable if controlling for the mediator variable reduces the variance of the dependent variable explained by the independent variable.

Mediation models were computed utilizing PROCESS v. 3.0 (*Hayes, 2012*). Since the product of the two variables is normally distributed only in large samples, we used bootstrapping (5000 samples) to construct bias-corrected and accelerated 95% confidence intervals (CIs) around a point estimate of the indirect effect (*Mackinnon et al., 2004*; *Preacher and Hayes, 2008*). This procedure tested the null hypothesis that the indirect path from the interaction term to the dependent variable via the mediator does not differ from 0. If 0 is not contained within the CIs computed by the bootstrapping procedure, the indirect effect is inferred to differ from 0 at p<0.05.

## Multiple step-wise linear regression analyses

We assessed the proportion of the variance of memory scores that was explained by structural and functional abnormalities across patients by entering measures of such abnormalities (that were not significantly correlated across patients) in a series of multiple step-wise linear regression analyses as independent variables (default alpha level of 0.05 for entry to model and 0.1 for removal).

## Acknowledgements

We are very grateful to the participants who took part in this study, as well as to Prof. Giovanna Zamboni (Università di Modena e Reggio Emilia) for granting us access to additional structural MRI datasets, Prof. Andy Lee (University of Toronto Scarborough) for his helpful feedback on an earlier version of this manuscript, and to Prof. Ricardo Insausti (University of Castilla-La Mancha) for his advice regarding manual segmentations.

SRI is supported by the Wellcome Trust (104079/Z/14/Z), the UCB-Oxford University Alliance, BMA Research Grants- Vera Down grant (2013) and Margaret Temple (2017), Epilepsy Research UK (P1201) and by the Fulbright UK-US commission (MS-SOCIETY research AWARD). The research was funded/supported by the National Institute for Health Research (NIHR) Oxford Biomedical Research Centre (BRC; The views expressed are those of the author(s) and not necessarily those of the NHS, the NIHR or the Department of Health).

CRB is supported by a Medical Research Council Clinician Scientist Fellowship (MR/K010395/1).

# Additional information

## Competing interests

Sarosh R Irani: is a coapplicant and receives royalties on patent application WO/2010/046716 (U.K. patent no., PCT/GB2009/051441) entitled 'Neurological Autoimmune Disorders'. The patent has been licensed to Euroimmun AG for the development of assays for LGI1 and other VGKC-complex antibodies. The other authors declare that no competing interests exist.

## Funding

| Funder | Grant reference number | Author |
|---|---|---|
| Wellcome Trust | 104079/Z/14/Z | Sarosh R Irani |
| UCB-Oxford University Alliance | | Sarosh R Irani |
| British Medical Association | Vera Down Grant (2013) | Sarosh R Irani |
| British Medical Association | Margaret Temple Grant (2017) | Sarosh R Irani |
| Epilepsy Research UK | P1201 | Sarosh R Irani |
| US-UK Fulbright Commission | MS-SOCIETY research AWARD | Sarosh R Irani |
| Medical Research Council | MR/K010395/1 | Christopher R Butler |

The funders had no role in study design, data collection and interpretation, or the decision to submit the work for publication.

## Author contributions

Georgios PD Argyropoulos, Conceptualization, Data curation, Formal analysis, Investigation, Visualization, Methodology, Writing—original draft, Project administration; Clare Loane, Conceptualization, Data curation, Formal analysis, Investigation, Methodology, Project administration; Adriana Roca-Fernandez, Data curation, Formal analysis, Methodology; Carmen Lage-Martinez, Data curation, Investigation, Methodology, Project administration; Oana Gurau, Investigation, Writing—original draft; Sarosh R Irani, Data curation, Writing—original draft; Christopher R Butler, Conceptualization, Supervision, Funding acquisition, Investigation, Methodology, Writing—original draft, Project administration

## Author ORCIDs

Georgios PD Argyropoulos (iD) https://orcid.org/0000-0001-8267-6861
Christopher R Butler (iD) https://orcid.org/0000-0002-7502-9284

## Ethics

Human subjects: All participants provided written informed consent according to the Declaration of Helsinki. Ethical approval was received from South Central Oxford Research Ethics Committee (REC no: 08/H0606/133).

## Decision letter and Author response

Decision letter https://doi.org/10.7554/eLife.46156.026
Author response https://doi.org/10.7554/eLife.46156.027

# Additional files

## Supplementary files

• Supplementary file 1. Supplementary tables. (**A**) Composite Memory scores for autoimmune LE patients and healthy controls; 'max = 18': the maximum score sum attainable for the autobiographical memory for childhood and early adulthood in the AMI (*Kopelman et al., 1989*), which does not

use age-scaled / standardized scores; p-corr: p values are adjusted for multiple testing using the Holm-Bonferroni sequential method (n=3); U: Mann-Whitney U; Wt: Welch's test; z: standardized age-scaled scores; *: Shapiro-Wilk test: p < 0.05. (**B**) Relationship of delay since symptom onset with structural and functional abnormalities. Given our hypothesis that broader network abnormalities unfold as a consequence of acute HPC atrophy (i.e. at a time-point after the acute focus of HPC damage), we also assessed the relationship of the delay between symptom onset and research participation with the extent of HPC and extra-HPC structural and functional abnormalities in a series of bivariate correlations. Of the structural/functional abnormalities (n=13) identified in our patient group, only inter-HPC rsFC decreased across patients as a function of the delay between symptom onset and research scan (rho=-0.58, p-corr=0.004). No other brain abnormalities showed this relationship, even at uncorrected levels (all rhos, |rho| ≤ 0.255; all ps, p-unc ≥ 0.122); bold: p-corr < 0.05; HPC: hippocampus; L, R: left, right (hemisphere); MPFC: medial prefrontal cortex; p-corr: p values are adjusted for multiple testing (n=13) using the Holm-Bonferroni sequential method; PCC: posterior cingulate cortex; PrCu: precuneus; rho: Spearmann's rank correlation coefficient; rsALFF: resting-state amplitude of low frequency fluctuations; rsFC: resting-state functional connectivity; OPTIMA: Oxford Project To Investigate Memory and Aging; MAP: Memory and Amnesia Project; TIV: total intracranial volume; z-res: volumes are residualized against age, sex, scan source (MAP, OPTIMA), and TIV; functional abnormalities are residualized against age and sex. (**C**) Correlation of memory scores with structural / functional abnormalities across patients [analysis-level correction for multiple testing (n=39)]; bold: p-corr < 0.05; GM: gray matter; HPC: hippocampus; L: left hemisphere; M: medial; MAP: Memory and Amnesia Project; MPFC: medial prefrontal cortex; OPTIMA: Oxford Project To Investigate Memory and Aging; PCC: posterior cingulate cortex; p-corr: p values of bivariate correlations are corrected for multiple testing using the Holm-Bonferroni sequential method of correction for the total number of correlations (n = 39); PMC: posteromedial cortex; PrCu: precuneus; rsALFF: amplitude of low frequency fluctuations; rsFC: resting-state functional connectivity; R: right hemisphere; TIV: total intracranial volume; VBM: voxel-based morphometry; z-res: Mean rsALFF and rsFC values are residualized for age and sex across participants; volumes are residualized against age, sex, scan source (OPTIMA, MAP) and TIV across participants. (**D**) Correlation of memory scores with structural / functional abnormalities across patients [score-level correction for multiple testing (n=13), separately for each of the different memory scores (n=6) examined]; bold: p-corr < 0.05; HPC: hippocampus; MAP: Memory and Amnesia Project; MPFC: medial prefrontal cortex; OPTIMA: Oxford Project To Investigate Memory and Aging; PCC: posterior cingulate cortex; p-corr: p values of bivariate correlations are corrected for multiple testing using the Holm-Bonferroni sequential method of correction for the number of different variables (n = 13) per memory score examined; PrCu: precuneus; rsALFF: amplitude of low frequency fluctuations; rsFC: resting-state functional connectivity; VBM: voxel-based morphometry; z-res: Mean rsALFF and rsFC values are residualized for age and sex across participants; volumes (derived from manual / automated segmentation or from the mean GM volume expressed by VBM clusters) were residualized against age, sex, TIV, and scan source (MAP, OPTIMA) across participants. (**E**) Relationship of memory scores with volumes of manually delineated HPC portions. *: verbal / visual recognition, visual recall composite scores: No significant correlations were observed between those scores and the volumes of the manually delineated left/right anterior/posterior HPC portions at uncorrected levels (all ps, p-unc > 0.05); **: verbal recall composite scores: A weak correlation was observed with the left anterior HPC at uncorrected levels (r = 0.351, p-unc = 0.033; rest of ps, p-unc ≥ 0.095). Since patients' rsALFF in the PCC did not correlate significantly with the volume of the manually delineated left anterior HPC (r = 0.333, p = 0.050), we entered these two factors as independent variables in a multiple step-wise linear regression, with verbal recall scores as the dependent variable. The regression terminated in a single step, with rsALFF in the PCC as the only predictor of patients' performance ($R^2$=0.34; β(z)=0.58; F=16.40, p < 0.0005); ***: visual forgetting scores: There was no clear evidence for a selective relationship of visual forgetting with anterior vs. posterior portions of the HPC (all rhos, 0.471≥ rho ≥ 0.332; all ps, 0.055 ≥ p-unc ≥ 0.005); ****: Correlations of remote autobiographical memory with volumes of manually delineated portions of the HPC (anterior / posterior right / left HPC) were observed at uncorrected levels (0.429 ≥ r ≥ 0.334; 0.066 ≥ p-unc ≥ 0.016). All four HPC portions correlated volumetrically with the left thalamus (right anterior HPC: r = 0.327, p-unc = 0.045; left anterior HPC: r = 0.427, p-unc = 0.007; right posterior HPC: r = 0.362, p-unc = 0.025; left posterior HPC: r = 0.360, p-unc = 0.026). A series of four partial correlational analyses demonstrated

that left thalamic volume correlated with remote autobiographical memory scores over and above the volume of each of those four manually delineated HPC portions (control variable: right anterior HPC: r = 0.489, p = 0.006; left anterior HPC: r = 0.482, p = 0.007; right posterior HPC: r = 0.505; p = 0.004; left posterior HPC: r = 0.509, p = 0.004); aHPC: anterior hippocampus; HPC: hippocampus; L: left hemisphere; MAP: Memory and Amnesia Project; OPTIMA: Oxford Project To Investigate Memory and Aging; pHPC: posterior hippocampus; p-unc: p values of bivariate correlations are presented at uncorrected levels for display purposes; R: right hemisphere; rho: Spearmann's rank correlation coefficient; r: Pearson's correlation coefficient; TIV: total intracranial volume; z: average of age-scaled standardized scores on neuropsychological tests of episodic memory; z-res: volumes for each HPC portion are residualized against age, sex, TIV, and scan source (MAP, OPTIMA) across participants. (F) 'Impaired' vs. 'Unimpaired' patients on visual forgetting: Structural/Functional abnormalities. Given that 17 of our patients reached ceiling scores in visual forgetting, we also dichotomized our patient group into two subgroups, those that attained ceiling scores (z=0.33), and those with lower scores (z<0.33). We therefore compared the two patient subgroups across the 13 structural and functional abnormalities identified above at whole-group level. Consistent with our correlational approach, the two patient subgroups differed only with respect to the volume of the right HPC, as expressed in manual volumetry (t = -3.32, p-corr = 0.027) and in the right HPC VBM cluster (t = -4.05, p-corr = 0.004; rest of ps, p-corr ≥ 0.143); We then iterated these comparisons with a series of one-way ANCOVAs, including patients' HADS scores for depression as a covariate of no interest. The results retained their significance (manually delineated right HPC volume: F = 9.96; p-corr = 0.048; right HPC VBM cluster: F = 14.94, p-corr = 0.007; rest of ps, p-corr ≥ 0.232); p-corr: significance values are corrected for multiple testing using the Holm-Bonferroni sequential method (*Holm, 1979*); t/Wt: comparison between the two subgroups across the 13 structural/functional abnormalities identified at group level for patients; F: these comparisons were iterated in the form of a series of univariate ANCOVAs, including patients' scores for depression (HADS) as between-subjects covariates; z-res: volumes are residualized against age, sex, scan source (MAP, OPTIMA), and TIV; functional abnormalities are residualized against age and sex; t: Student t-test; Wt: Welch t-test; SD: standard deviation; bold: p-corr < 0.05; HADS: Hospital Anxiety and Depression Scale (*Zigmond and Snaith, 1983*); ANCOVA: analysis of covariance; OPTIMA: Oxford Project To Investigate Memory and Aging; MAP: Memory and Amnesia Project. (G) Comparison of treated vs. untreated patients on memory scores and structural/functional brain abnormalities. We investigated whether patients that had been treated with immunosuppressive therapy (n=31) showed less pronounced memory impairment and brain abnormalities as compared with those that had not received treatment (n=7), in order to ensure that the structure/function-behavior relationships disclosed across all 38 patients were not driven by the subgroup of patients that had not received such treatment. Patients who had not received such treatment scored lower than the rest of the patients on anterograde retrieval, but not on visual forgetting, or remote autobiographical memory. Nevertheless, the two patient subgroups did not differ with respect to the extent of structural or functional brain abnormalities that we observed above for the entire group of patients (all ps, p-corr ≥ 0.832), even at uncorrected levels (all ps, p-unc > 0.06); p-corr: Holm-Bonferroni (*Holm, 1979*) correction applied separately for the number of memory scores (n=3) and the number of structural/functional abnormalities (n=13); bold: p-corr < 0.05; z: average of age-scaled standardized scores on neuropsychological tests of episodic memory; z-res: volumes are residualized against age, sex, scan source (MAP, OPTIMA), and TIV; functional abnormalities are residualized against age and sex. (H) Correlation of anterograde retrieval scores with structural / functional abnormalities across patients that had received immunosuppressive therapy. Given the more pronounced impairment on anterograde retrieval of patients that had not received immunosuppressive therapy, we sought to determine whether the relationship of this composite score with reduced rsALFF in the PCC held when the analysis was confined to the 31 patients that had received immunosuppressive therapy. Indeed, the relationship between reduced rsALFF in the PCC and impaired anterograde retrieval retained its significance across this patient subgroup; L, R: left, right (hemisphere); HPC: hippocampus; MPFC: medial prefrontal cortex; p-corr: p values are adjusted for multiple testing using the Holm-Bonferroni sequential method; bold: p-corr < 0.05; rsALFF: resting-state amplitude of low frequency fluctuations; rsFC: resting-state functional connectivity; PrCu: precuneus; PCC: posterior cingulate cortex; z-res: volumes are residualized against age, sex, scan source (MAP, OPTIMA), and TIV; functional abnormalities are residualized against age and sex. (I) Participants - data availability. Numbers of

healthy controls and patients that underwent structural, functional MRI, and neuropsychological assessment. *: While the healthy controls whose structural MRI datasets were added from the OPTIMA project had not been assessed with our laboratory's neuropsychological battery, they had been assessed with tests measuring overall cognitive impairment [Mini-Mental State Examination – MMSE; (*Folstein et al., 1975*)]. Expectedly, these scores indicated that none of those healthy controls had any apparent cognitive impairment [mean = 29.74, SD = 0.56, min = 28, well above widely accepted cut-offs, for example (*Aevarsson and Skoog, 2000*; *Di Carlo et al., 2002*)]. OPTIMA: Oxford Project To Investigate Memory and Aging; MAP: Memory and Amnesia Project; (rsf)MRI: (resting-state functional) Magnetic Resonance Imaging; n: number of participants.

DOI: https://doi.org/10.7554/eLife.46156.023

• Transparent reporting form

DOI: https://doi.org/10.7554/eLife.46156.024

## Data availability

A source data file has been provided for the plots in Figure 2—figure supplement 1 and Figures 3-8. The participants of our study had not been asked to consent for their anonymized data to be publicly shared and be made freely available. Therefore, these data are available through a request to Dr Christopher Butler and would need to be approved by an ethics committee. Information relating to the 32 MRI datasets previously collected and made available via the Oxford Project To Investigate Memory and Aging can be found at https://www.ndcn.ox.ac.uk/research/centre-prevention-stroke-dementia/resources/optima-oxford-project-to-investigate-memory-and-ageing and in Zamboni et al. (2013) (https://doi.org/10.1016/j.biopsych.2013.04.015). Requests to access should be made to Dr. Christopher Butler.

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
