## [Decision Letter]

Thank you for sending your article entitled "Network-wide abnormalities explain memory variability in hippocampal amnesia" for peer review at *eLife*. Your article is being evaluated by Laura Colgin as the Senior Editor, a Reviewing Editor, and three reviewers.

Given the list of essential revisions, including major concerns about key analyses, the editors and reviewers invite you to respond with an action plan and timetable for the completion of the additional work. This revision plan should detail how you propose to address the reviewers' major concerns.

*Reviewer #1:*

This paper examines the relationship between (1) hippocampal (HPC) volume measured by MRI and (2) memory deficits in standard neuropsychological tests on a group of N=38 patients with autoimmune limbic encephalitis, in particular whether this relationship is mediated by (3) the effects of HPC damage on other measures of brain structure (eg thalamus) or brain function (e.g. reduced resting-state "activity" in medial parietal cortex or reduced fMRI connectivity between left and right HPC). The results are potentially interesting and important because, although a number of other recent studies have used MRI and fMRI to examine effects of HPC lesions on other parts of the brain, in small groups of 2-6 patients, this paper is able to go further, by virtue of the much larger sample size, in comparing the relationship between those brain changes and a range of memory measures. The authors' main claim is that the majority of those memory measures are better explained by functional changes, or thalamic structural changes, than the conventional (implicit) assumption that HPC structural change is the main determinant. However, the paper at the moment is poorly structured, making it very difficult to understand the rationale for each analysis and the specific groups used, and the statistical interpretations need refinement.

1) A lot of the conclusions are based on mediation models in which the relationship between HPC volume (A) and a memory measure (C) is mediated by some other brain measure (e.g. functional connectivity, or thalamic volume (B)). Unfortunately, because mediation models (when implemented as path models) are fully saturated (particularly when all variables are observed quantities with measurement error, as here, rather than fixed effects that are manipulated), statistical methods cannot be used to test whether one model is any better than another, so the models need to be justified on theoretical grounds. In the present cases, while I accept that memory measures are most naturally the outcome (C), it seems plausible to me that one could swap A and B (e.g., claim that HPC volume mediates the relationship between thalamic volume and cognition). I think the authors should explicitly discuss this caveat (i.e., that their mediation results can only be interpreted under assumption that the disease primarily causes HPC damage, and this in turn causes the damage/functional disruption in other brain regions, rather than the alternative possibility that the disease causes, e.g., thalamic damage, but the effect of that damage on cognition is mediated by secondary HPC damage). This should not only be discussed at length in the Discussion, but also stated in the Introduction to explain why mediation analyses performed at all (e.g., "On the assumption that the disease causes HPC damage and that this then causes disruption in the extended hippocampal network, we performed mediation analysis to see whether disruption elsewhere affected the relationship between HPC damage and memory measures…etc." or the like". In other words, the Introduction should not just give the three "predictions" (which are actually more like open-ended hypotheses, since nobody is cited as explicitly claiming previously that memory problems arise from somewhere other than hippocampus), but also a summary of the (complicated) statistical approach taken to address these hypotheses.

2) Why were HPC volumes compared in terms of a one-sample T-test on Z-scores for each patient versus a matched group of controls? Why not simply perform a two-sample T-test comparing the two groups, after matching control group for mean and spread of age, sex, etc, and/or covarying out these variables? That way, the authors could report a more generalizable effect size in terms of mean and standard-error of volume estimates for each group (e.g., in Table 2, from which one can always derive a group Z-score or a% volume loss if one wants), in order to compare with other studies, rather than reporting a measure that is specific to the present control group (which is not particularly large anyway, ie likely to result in noisy Z-scores). (Also, with the number of controls specified, doesn't this patient-matching procedure result in the same control participant being used as a control for multiple patients? If so, does this further reduce the generalisability of the Z-scores?)

3) In general, although it is conventional to separate Results section from Materials and methods section of a paper, the results here are so terse, and the methods (design, analysis) so complicated, that I think it would really help readability of the paper if the Results section contained more information about the rationale and details (e.g., participant samples) used in each analysis. It would also help to have a summary to pre-empt how the results are organised (e.g., first we looked at structural changes, then we used regions from these to relate to cognition, etc… i.e., a roadmap of where results are heading). Indeed, since different analyses used different subsets of patients and controls, I wonder whether a flow-chart could be added to explain which participants contributed to which analysis?

4) No rationale is given (e.g., in Introduction) for why the authors chose to measure resting-state functional "activity" (rsALFF), or specific aspects of functional connectivity. It is fine that the specific regions/connections are selected from the data, ie after a voxel-wise search for mean difference between patients vs controls (since all subsequent correlations are done on patient group only, so unbiased by this prior selection), but why was resting-state "activity" examined at all? Also, given prior studies claiming certain functional connections/networks are disrupted by HPC damage, why did the authors not examine these specifically? Or why did they not focus on the regions in the "extended hippocampal network" they mention in the Introduction, reducing the multiple comparison problem?

5) Personally, I think the most interesting results (given above concerns about mediation analyses) are when a variable like rsFC correlates better with a memory measure than does HPC volume. While it is interesting to know further that when a variable like rsFC is added to a stepwise regression together with HPC volume, only the former remains significant, one cannot conclude from absence of evidence that there is no effect of HPC on, e.g. verbal anterograde memory. Would it be possible to explicitly compare the regression slopes in a model with both variables (after they have been Z-scored so Betas comparable), that is to test whether there is significant positive evidence that rsFC is more strongly related to a memory score than is HPC volume?

6) In general, there are a large number of statistical tests performed (e.g., even when just considering the 6 different types of memory score). Even if the authors continue to report all results at p<.05 uncorrected, it might be worth summarising at the end of the Results section those for which they are most confident (i.e., that would survive such correction).

*Reviewer #2:*

Summary: This study reports a novel and unique investigation into the brain systems that account for memory impairments in a group of individuals (n=38) who have developed hippocampal amnesia as a result of autoimmune limbic encephalitis. This study provides one of largest investigations into the causal contribution of the hippocampal formation toward memory function. The authors completed a set of memory tests, including tests of both retrograde and anterograde memory, as well as tests of remote episodic/autobiographical memory. Through this neuropsychological battery, both verbal and visual memory was tested, and different aspects of memory were probed (e.g., recall, recognition performance, forgetting). In addition, brain imaging was performed to obtain estimates of structural integrity of brain regions in the extended hippocampal network as well as network-wide functional connectivity. Memory scores were then correlated with the brain structural and functional measures, and these analyses revealed that many of the relationships between hippocampal volume and memory were mediated by wider network abnormalities. These findings provide important linkages with other work in the literature connecting hippocampal compromise to structural and functional integrity of the thalamus and the posterior parietal cortex.

In general, I found the paper well written, and that the methods and statistical analyses were conducted rigorously. There are a few methodological details that should be stated more clearly in the manuscript, which I have detailed below. I have some other comments and questions regarding the pattern of anatomical changes observed.

1) The finding that thalamic grey matter is reduced in these amnesic individuals is consistent with the literature on developmental amnesia (e.g. Dzieciol et al., 2017). Were the mammillary bodies examined in this dataset? It would be interesting to know if individuals who sustain hippocampal compromise during adulthood also develop atrophy within these structures. Alternatively, it would also be interesting to know if thalamic atrophy might occur in the absence of changes to the mammillary bodies, given that in the macaque and human there appear to be direct fornical projections from the hippocampus/subiculum to the anterior thalamus (Bubb et al., 2017).

2) It is stated that inflammatory pathology would be 'expected to spread to adjacent areas, rather than to remote regions in the extended HPC network.' Is there empirical evidence for this assertion? Or, conversely, might there be evidence that inflammatory pathology could spread from the hippocampus to the thalamus via the fornix?

3) A bit more information about the different types of functional connectivity is needed. From my read of the paper, two types of functional connectivity were performed: resting state functional connectivity (rsFC) and resting-state amplitude of low frequency fluctuations (rsALFF). Why were these two methods used and what can one method offer that the other method cannot? This should be made clearer either in the Materials and methods section or earlier in the manuscript.

4) Were mediation analyses performed for each of the correlations and only reported for the models in which either the thalamic or connectivity patterns significantly mediate the relationship between hippocampal volume and memory? In some parts of the paper these mediation analyses seem hypothesis driven (e.g. subsection “Anterograde memory: Visual Recognition”) whereas in other parts of the paper I wasn't sure if these mediation analyses were hypothesized a priori (e.g. subsection “Remote Autobiographical Memory”). This issue should be made clarified.

*Reviewer #3:*

This is an important and timely study of memory processing in patients with autoimmune limbic encephalitis who have relatively selective damage to the hippocampus. Given how rare it is to find individuals with focal damage to the hippocampus, the group of 38 patients that are described in the study is extremely large. The cohort is well-characterised from both neuroimaging and neuropsychological perspectives. The main findings are that, while the patients have relatively focal volume loss within the hippocampus, the relationships between hippocampal volume and memory performance are usually mediated by relationships with other brain structures: notably functional connectivity with posterior midline regions in the case of anterograde memory and grey matter volume in the thalamus in the case of retrograde memory. I think these findings will make a useful contribution to the field of memory research, but I have a few issues that I feel the authors should address.

In the volumetric analyses, the authors seem to be testing whether the patient group as a whole has volume loss greater than 1.5 SDs below the mean of the controls. If correct, this seems to be extremely conservative and rather unusual. I would have expected a more direct comparison between the volumes of the patients and controls – with indications of the effect size of the difference. This is the approach done with the neuropsychological tests. It might help if the degree of volume loss is expressed as a percentage of the controls, with statistics relating to the direct comparison of the patients and controls. Perhaps I am missing something here?

The authors correlate MRI measures with composite measures of performance based on averaging across tests of anterograde memory in which patients showed impairment. This seems an inadequate justification for choosing which tests contribute to the composite measures. It is interesting that the patients are unimpaired on tests of face recognition memory, and as the authors point out, this is in keeping with other findings (e.g. Taylor et al., 2007; Bird and Burgess, 2008; Smith et al., 2014; Bird, 2017). This would constitute a reasonable justification for not including face recognition memory in a composite visual recognition memory measure in patients with hippocampal damage. However, the selection of measures based on performance alone is biased.

In the description of the methods for "Resting-state Hemodynamic activity: rsALFF" the authors define RsALFF as the total power within the frequency rage (0.01-0.1 Hz). However, in the pre-processing section they say that they apply a temporal band pass filter of 0.01-0.1 Hz. To me, this would appear to affect the signal that they want to analyse. Could the authors justify their procedure and confirm if this procedure is used by other groups reporting the results of RsALFF analyses?

The study and its results are situated with respect to neural dysfunction at sites separated from the hippocampus. While this is very likely to be the cause of volume loss in the thalamus, it is not necessarily the case for the resting state fMRI results. It is possible that the patients engage different cognitive processes during the resting state scan compared to the controls – and therefore the differences in the engagement of the anterior and posterior midline structures reflects this. These regions are strongly implicated in "mind-wandering" – following undirected trains of thought. There is also evidence that patients with hippocampal damage differ markedly in the form and content of their mind-wandering compared with healthy adults (McCormick et al., 2018). It is also possible that the patients simply forgot what they were doing in the scanner, and spent more time attending to, e.g., the noise of the scanner than the controls did. My point is that functional differences between the patients and controls could reflect the BOLD engagement in different cognitive processes rather than more fundamental differences in neuro/vascular functioning in these regions per se. I would encourage the authors to discuss this issue.

---

## [Author Response]

Reviewer #1:

This paper examines the relationship between (1) hippocampal (HPC) volume measured by MRI and (2) memory deficits in standard neuropsychological tests on a group of N=38 patients with autoimmune limbic encephalitis, in particular whether this relationship is mediated by (3) the effects of HPC damage on other measures of brain structure (e.g. thalamus) or brain function (e.g. reduced resting-state "activity" in medial parietal cortex or reduced fMRI connectivity between left and right HPC). The results are potentially interesting and important because, although a number of other recent studies have used MRI and fMRI to examine effects of HPC lesions on other parts of the brain, in small groups of 2-6 patients, this paper is able to go further, by virtue of the much larger sample size, in comparing the relationship between those brain changes and a range of memory measures. The authors' main claim is that the majority of those memory measures are better explained by functional changes, or thalamic structural changes, than the conventional (implicit) assumption that HPC structural change is the main determinant. However, the paper at the moment is poorly structured, making it very difficult to understand the rationale for each analysis and the specific groups used, and the statistical interpretations need refinement.

We would like to thank our reviewer for the encouraging feedback. In order to improve the structure of the paper, we have amended the manuscript in line with the recommendations provided in the reviewer’s subsequent comments.

1) A lot of the conclusions are based on mediation models in which the relationship between HPC volume (A) and a memory measure (C) is mediated by some other brain measure (e.g. functional connectivity, or thalamic volume (B)). Unfortunately, because mediation models (when implemented as path models) are fully saturated (particularly when all variables are observed quantities with measurement error, as here, rather than fixed effects that are manipulated), statistical methods cannot be used to test whether one model is any better than another, so the models need to be justified on theoretical grounds. In the present cases, while I accept that memory measures are most naturally the outcome (C), it seems plausible to me that one could swap A and B (e.g., claim that HPC volume mediates the relationship between thalamic volume and cognition). I think the authors should explicitly discuss this caveat (i.e., that their mediation results can only be interpreted under assumption that the disease primarily causes HPC damage, and this in turn causes the damage/functional disruption in other brain regions, rather than the alternative possibility that the disease causes, e.g. thalamic damage, but the effect of that damage on cognition is mediated by secondary HPC damage). This should not only be discussed at length in the Discussion, but also stated in the Introduction to explain why mediation analyses performed at all (e.g., "On the assumption that the disease causes HPC damage and that this then causes disruption in the extended hippocampal network, we performed mediation analysis to see whether disruption elsewhere affected the relationship between HPC damage and memory measures…etc." or the like". In other words, the Introduction should not just give the three "predictions" (which are actually more like open-ended hypotheses, since nobody is cited as explicitly claiming previously that memory problems arise from somewhere other than hippocampus), but also a summary of the (complicated) statistical approach taken to address these hypotheses.

We agree with the reviewer that the manuscript would be improved by explicit discussion of the assumptions of the mediation analysis. The specific assumption that the reviewer refers to – i.e. that the disease primarily causes HPC damage and that this in turn leads to disruption in other brain regions, rather than the alternative possibility that the disease causes, e.g., thalamic damage, but the effect of that damage on cognition is mediated by secondary HPC damage – is, we believe, well founded for the reasons that we have indicated (and now further highlighted) in the revised Introduction and Discussion section, as follows:

Introduction: ‘In its acute phase, this highly consistent clinical syndrome classically causes focal HPC damage, seen on MRI as high T2 signal (Finke et al., 2017; Kotsenas et al., 2014; Loane et al., 2019; Malter et al., 2014). Although most patients respond well to early immunosuppressive therapy (Finke et al., 2017; Irani et al., 2011; Thompson et al., 2018), some subsequently develop HPC atrophy and persistent cognitive impairment, the most prominent aspect of which is anterograde amnesia (Butler et al., 2014; Finke et al., 2017; Irani et al., 2013; Loane et al., 2019; Malter et al., 2014). Autoimmune LE patients are often included in studies of HPC amnesia (Hassabis et al., 2007; Henson et al., 2017, 2016; Maguire et al., 2006). Post-mortem studies demonstrate focal HPC damage (Khan et al., 2009; Park et al., 2007), contrasting with the wider damage typically found following other encephalitides [e.g. herpes simplex encephalitis (Damasio and Van Hoesen, 1985; Gitelman et al., 2001)] or global ischemia/anoxia (Huang and Castillo, 2008), conditions that have also been used as models of human HPC function [for discussion, see (Squire and Zola, 1996)]. Animal models have also shown focal limbic involvement (HPC, amygdala) (Tröscher et al., 2017). Patients’ varying degrees of residual symptom severity offered us the opportunity to examine the relationships between cognition and brain structure and function.’

Discussion section: ‘[…] acute inflammatory changes appeared confined to the HPC in the vast majority of our patients, while no such changes were found in the thalamus. Moreover, clinical T2-weighted MRIs in other autoimmune LE cohorts (Finke et al., 2017; Kotsenas et al., 2014; Malter et al., 2014), and the results of post-mortem studies have also demonstrated focal, acute HPC pathology in autoimmune LE patients (Khan et al., 2009; Park et al., 2007) and animal models (Tröscher et al., 2017). Our interpretation of these data is, therefore, that extra-HPC structures within the extended HPC system (thalamus, PCC) are affected as a secondary consequence of HPC damage, rather than due to the primary pathology. We also note that extra-HPC damage is equally if not more likely in other conditions associated with HPC amnesia, such as cases of ischemia/anoxia (Huang and Castillo, 2008).’

We have explicitly justified our reasons for using mediation analysis and discussed the assumptions that underlie these analyses at the end of our revised Introduction. To address the reviewer’s additional comment that nobody is cited as explicitly claiming previously that memory problems arise from somewhere other than the HPC and that therefore the questions presented in the end of our Introduction are more like ‘open-ended questions’, we have cited recently published papers that highlight the significance of broader networks in episodic memory and propose that episodic memory impairment results from broader network-wide disruption triggered by HPC damage [see for instance (Inhoff and Ranganath, 2017), where this proposal is explicitly made].

‘We predicted that: (i) HPC damage (documented to occur focally in autoimmune LE) would be accompanied by remote abnormalities in hubs of the extended HPC system, including the HPC-diencephalic-cingulate network (Aggleton, 2014; Bubb et al., 2017a); (ii) variation in the extent of these network-wide abnormalities across patients would explain variability in memory impairment over and above HPC atrophy; iii) Moreover, on our assumption that HPC damage leads to broader network disruption, we predicted that relationships between HPC atrophy and memory impairment would be mediated by this broader network disruption. This prediction is in line with recent functional imaging and lesion studies that emphasize the importance of broader networks in episodic memory (Cook et al., 2015; Henson et al., 2016; Inhoff and Ranganath, 2017; Schedlbauer et al., 2015; Watrous et al., 2013); iv) taking these abnormalities into account would enable us to identify which particular aspects of memory impairment are a direct function of HPC atrophy.’

2) Why were HPC volumes compared in terms of a one-sample T-test on Z-scores for each patient versus a matched group of controls? Why not simply perform a two-sample T-test comparing the two groups, after matching control group for mean and spread of age, sex, etc, and/or covarying out these variables? That way, the authors could report a more generalizable effect size in terms of mean and standard-error of volume estimates for each group (e.g., in Table 2, from which one can always derive a group Z-score or a% volume loss if one wants), in order to compare with other studies, rather than reporting a measure that is specific to the present control group (which is not particularly large anyway, ie likely to result in noisy Z-scores). (Also, with the number of controls specified, doesn't this patient-matching procedure result in the same control participant being used as a control for multiple patients? If so, does this further reduce the generalisability of the Z-scores?)

Our original approach was adopted in keeping with standard practice in single-case and case-series studies of patients with HPC damage, whereby HPC volumes of individual patients are shown to fall below the traditionally employed cut-off point of -1.5 SD from the mean of age-matched healthy controls, in the face of other MTL (and subcortical) structures showing volumes over -1.5 SD from the mean of their matched healthy controls. Nevertheless, we agree that, since the number of healthy controls is limited and that some participants are used as age-matched controls for multiple patients, this method of quantifying MTL and subcortical volumes may not be ideal. Moreover, we agree that the advantages of the approach the reviewer has recommended may give less noisy estimates of volume reduction, and that the effect sizes would be more generalisable for comparison with other studies. A similar point was raised by reviewer 3. We have now adopted the method of quantifying MTL and other subcortical volumes that the reviewer recommends. We have iterated the analyses of structure/function-behavior relationships, after first regressing MTL and subcortical volumes against age, sex, and TIV (as already done in the case of the mean values of VBM clusters in the original manuscript), as well as the project in which the structural datasets were acquired (Memory and Amnesia Project, Oxford Project To Investigate Memory and Aging), as recommended in another point below. This has been highlighted in several instances, as, for instance, in the Materials and methods section:

Volumes derived from manual volumetry: ‘We compared healthy controls and patients on the volumes of all structures in a series of ANCOVAs, using age, sex, scan source [MAP (Memory and Amnesia Project), or OPTIMA (Oxford Project To Investigate Memory and Aging)], and total intra-cranial volume (TIV; derived from the unified segmentation procedure in SPM12; see below) as between-subjects covariates. Volumes that were reduced in our patient group were residualized against these four variables (z-res) and entered in a series of bivariate correlations with patients’ memory scores.’

Volumes derived from VBM: ‘The mean GM volumes of VBM clusters were extracted using the Marsbar toolbox in SPM12 (Brett et al., 2002), were residualized across participants against age, sex, scan source (MAP, OPTIMA), and TIV, and were used in a series of bivariate correlations with memory scores across patients.’

3) In general, although it is conventional to separate Results section from Materials and methods section of a paper, the results here are so terse, and the methods (design, analysis) so complicated, that I think it would really help readability of the paper if the Results section contained more information about the rationale and details (e.g., participant samples) used in each analysis. It would also help to have a summary to pre-empt how the results are organised (e.g., first we looked at structural changes, then we used regions from these to relate to cognition, etc… i.e., a roadmap of where results are heading). Indeed, since different analyses used different subsets of patients and controls, I wonder whether a flow-chart could be added to explain which participants contributed to which analysis?

We agree that the reviewer’s suggested changes to the structure of the manuscript would enhance readability. We have now added more information in the Results section about the rationale and details used in each analysis, as follows:

Our analysis approach is summarized in Figure 1. We first (1) identified cognitive deficits by comparing the performance of patients and healthy controls on a broad range of neuropsychological tests. We then (2) identified regions in which patients showed structural and (3) functional abnormalities relative to healthy controls. We then (4) examined the relationship between structural / functional brain abnormalities and memory impairment across patients.

We have also added more information about methods at other places throughout the Results section, especially with respect to imaging.

As the reviewer recommended, we have also included a figure illustrating how the results are organised and which participants contributed to which analysis (Figure 1) (please also consult response below regarding the number of healthy controls used in the different analyses).

4) No rationale is given (e.g., in Introduction) for why the authors chose to measure resting-state functional "activity" (rsALFF), or specific aspects of functional connectivity. It is fine that the specific regions/connections are selected from the data, ie after a voxel-wise search for mean difference between patients vs controls (since all subsequent correlations are done on patient group only, so unbiased by this prior selection), but why was resting-state "activity" examined at all? Also, given prior studies claiming certain functional connections/networks are disrupted by HPC damage, why did the authors not examine these specifically? Or why did they not focus on the regions in the "extended hippocampal network" they mention in the Introduction, reducing the multiple comparison problem?

We have now taken the following steps in response to the reviewer’s point.

1) We have elaborated on the reasons for which we examined resting-state functional activity (in the form of low-frequency oscillations), in both the Introduction and the Materials and methods sections:

Introduction: ‘Using resting-state fMRI (rsfMRI), we also identified resting-state functional abnormalities in patients with respect to both segregation and integration, i.e. in terms of hemodynamic activity in local regions (resting-state amplitude of low-frequency fluctuations; rsALFF) and functional connectivity between regions (resting-state functional connectivity; rsFC).’

Materials and methods section: ‘We explored resting-state functional abnormalities in patients with respect to both segregation and integration, i.e. in terms of hemodynamic activity in local regions and functional connectivity between regions. While the majority of rsfMRI studies investigate the correlations between brain areas from the perspective of integration, i.e. ‘resting-state functional connectivity’ (rsFC) (providing holistic information on sets of interacting brain regions) and the abnormal integration between brain areas in patient groups relative to healthy controls, these approaches do not directly provide information on the specific brain regions that show abnormalities in patients (in the form of the amplitude of spontaneous brain activity). The latter are reliably indexed by the resting-state amplitude of low-frequency fluctuation (rsALFF) of the rsfMRI signal, and help us identify the specific brain regions of abnormal spontaneous activity (Zang et al., 2007; Zou et al., 2008), similar to resting-state CBF and glucose metabolic rate in PET studies. For the above reasons, the combined application of these two methods (rsFC and rsALFF) has been held to provide more information than either method alone (Lv et al., 2018). We used data-driven approaches in both.’

2) We have also now emphasized the advantages of our approach in the investigation of rsFC in our Materials and methods section:

‘Instead of selecting seed/target ROIs or networks in an a priori fashion for our rsFC analyses, we chose to capitalize on the size of our patient cohort and use a PCA-based approach, commonly referred to as ‘connectome-MVPA’ (‘multivariate pattern analysis’) [e.g. (Arnold Anteraper et al., 2019; Flodin et al., 2016; Kazumata et al., 2017; Thompson et al., 2016; Whitfield-Gabrieli et al., 2016; Yankouskaya et al., 2017); implemented in the Conn toolbox (Whitfield-Gabrieli and Nieto-Castanon, 2012) https://sites.google.com/view/conn/measures/networks-voxel-level], in order to identify seed regions for post-hoc seed-to-voxel connectivity analyses in a data-driven fashion. This method has been extensively used to identify the regions in which groups of patients differ significantly from healthy controls with respect to their rsFC with the rest of the brain (e.g. (Flodin et al., 2016; Kazumata et al., 2017; Thompson et al., 2016)). As outlined in other papers (e.g. (Whitfield-Gabrieli et al., 2016)), the strength of this approach is the use of a massive data set (connectivity between all pairs of recorded voxels) for the purposes of identifying the most reliable difference in rsFC across the whole brain between groups of participants. In other words, it is an agnostic, unbiased approach used to define data-driven regions of interest (seeds) prior to performing a post-hoc analysis on the seeds to analyse brain connectivity patterns. This PCA-based method is more reproducible than conventional seed-based approaches (Song et al., 2016). As this approach is both data-driven and conducted across the whole brain, it is strongly preferable to approaches examining a priori defined networks or seed/target regions of interest, commonly employed by studies of HPC amnesia with substantially smaller sample sizes.

In particular, the ‘connectome-MVPA’ method assesses the multivariate pattern of pairwise connections between voxels across the whole brain by performing a PCA separately for each voxel that characterizes its rsFC with the rest of the brain, in two steps. In the first step, separately for each participant, a default number (n=64) of Singular Value Decomposition (SVD) components characterizing each participant’s voxel-to-voxel correlation matrix is retained. The resulting component scores are stored as first-level voxel-to-voxel covariance matrices for each participant. In the second step, a low-dimensional representation of the entire pattern of seed-based correlations between this voxel and the rest of the brain is derived for each voxel across participants by retaining a certain number of principal components that explain most of the variance of the connectivity matrix. In our study, we retained the first 7 principal components, in keeping with a conventionally used conservative 1:10 ratio between the number of components extracted and the number of subjects (n=67). These 7 resulting component score volumes best represented the whole-brain connectivity pattern for each participant, explaining the maximum inter-subject variability. They were simultaneously included in a second-level analysis F-test at group-level (an omnibus test, equivalent to seed-level F-tests in ROI-to-ROI analyses of rsFC), testing for clusters that differ between healthy controls and LE patients with respect to whole-brain connectivity, as represented by the PCA component volumes, while also including age and sex as between subjects covariates. This method was then followed by post-hoc analyses to determine specific connectivity patterns in the data.’

5) Personally, I think the most interesting results (given above concerns about mediation analyses) are when a variable like rsFC correlates better with a memory measure than does HPC volume. While it is interesting to know further that when a variable like rsFC is added to a stepwise regression together with HPC volume, only the former remains significant, one cannot conclude from absence of evidence that there is no effect of HPC on, eg, verbal anterograde memory. Would it be possible to explicitly compare the regression slopes in a model with both variables (after they have been Z-scored so Betas comparable), that is to test whether there is significant positive evidence that rsFC is more strongly related to a memory score than is HPC volume?

We wish to highlight the following aspects of the analyses reported in the original version of our manuscript:

i) As clarified in response to point 1 above, there is very strong evidence supporting the assumption underlying our mediation analyses, namely, that the disease primarily causes HPC damage, and that this in turn causes the damage/functional disruption in other brain regions, rather than the alternative possibility that the disease causes, e.g., thalamic damage, but the effect of that damage on cognition is mediated by secondary HPC damage); we therefore strongly believe that our mediation analyses should be retained in the manuscript. Moreover, this has been highlighted in our revised manuscript (Introduction and Discussion section – see response to point 1 above).

ii) Furthermore, multiple stepwise regressions cannot be employed for every composite memory score, given that there is multicollinearity among the independent variables of interest (e.g. HPC volumes, thalamic volumes, and rsALFF in the PCC strongly correlate across patients). This was not an issue in the case of verbal recognition memory scores, where inter-HPC rsFC did not correlate across patients with HPC volumes.

iii) Comparisons between regression slopes in a model including both HPC volumes and other structural abnormalities as independent variables [such as the approach based on (Cumming, 2009), whereby two point estimates are likely statistically significantly different from each other, when the corresponding 95% confidence intervals overlap by not more than 50%] also rely on multiple linear regression, and, since the independent variables here are strongly correlated, the issue of multi-collinearity is evident.

iv) Moreover, we have consistently avoided claiming that memory impairment correlated ‘better’ or ‘more strongly’ with extra-HPC abnormalities as compared with HPC volumes. In the original version of our manuscript, we had shown that: (a) the correlations of memory scores with extra-HPC abnormalities were significant, whereas those with HPC abnormalities were not (after correction for multiple testing); (b) the effects of HPC abnormalities on memory scores are mediated by those correlative extra-HPC abnormalities (mediation analyses); (c) moreover, in the Materials and methods section and Results section of this revised version of the manuscript, we have now introduced a series of partial correlations, demonstrating that memory scores correlated with these extra-HPC abnormalities across patients over and above the correlative HPC atrophy. This is something that we also now highlight in our Discussion section, as follows:

‘At uncorrected levels, HPC volumes correlated with both recall and recognition memory scores. Crucially, however, the reduction in PCC rsALFF correlated with verbal and visual recall and visual recognition over and above HPC atrophy and, moreover, fully mediated the effects of HPC atrophy on these aspects of episodic memory. This finding demonstrates that the HPC role in recall and recognition cannot be examined in isolation from remote effects of HPC damage within the extended HPC system.’

6) In general, there are a large number of statistical tests performed (e.g., even when just considering the 6 different types of memory score). Even if the authors continue to report all results at p<.05 uncorrected, it might be worth summarising at the end of the Results those for which they are most confident (i.e., that would survive such correction).

In our original manuscript, we had applied Holm-Bonferroni correction (‘p-corr’) for multiple correlations of structural/functional abnormalities with memory impairment, applied separately for each of the 6 composite scores examined (verbal recognition, visual recognition, verbal recall, visual recall, visual forgetting, remote autobiographical memory). We only considered correlations at uncorrected levels (‘p-unc’) when they pertained to HPC volumes, in order to highlight the fact that, even when the aprioristic approach focusing on HPC atrophy is adopted, the effects of HPC volume reduction on episodic memory are fully mediated by correlative abnormalities in other regions of the extended HPC system (thalamic atrophy, reduced rsALFF in the PCC; except for visual forgetting). In the Results section, we have now highlighted the fact that correction for multiple correlations was applied. Moreover, in order to address the reviewer’s concern that there was no additional correction applied for the number of the 6 different composite scores (verbal recognition, visual recognition, verbal recall, visual recall, visual forgetting, remote autobiographical memory), our revised Results section now starts with an analysis that involves correction for multiple testing for the total number of correlational analyses, as below:

‘We first applied a stringent correction for multiple testing for the total number of correlations conducted (n=39) between the brain abnormalities identified (n=13; right/left HPC volumes and right entorhinal cortical volumes, based on manual delineation; left thalamic volumes, based on automated segmentation; anterior-mediodorsal and right dorsolateral thalamic volumes and right/left HPC volumes, expressed by the VBM clusters above; reduced rsALFF in the precuneus and the PCC; reduced rsFC between the right HPC and the left HPC, the medial prefrontal cortex, and the precuneus) and the three composite memory scores (n=3; anterograde retrieval; anterograde retention; remote autobiographical memory). Three correlations survived correction for multiple tests: i) Anterograde retrieval scores correlated across patients with their reduced rsALFF in the PCC (r = 0.551; p-corr = 0.024); ii) Anterograde retention (i.e. ‘forgetting’) scores correlated with patients’ reduced right HPC volume (VBM cluster; rho = 0.556, p-corr = 0.024; manually delineated right HPC volume: rho = 0.508, p-corr = 0.079); iii) Remote autobiographical memory scores correlated with patients’ reduced volume in the left thalamus (r = 0.558; p-corr = 0.041; rest of ps, p-corr ≥ 0.105; Supplementary Table 3).

Given the striking lack of correlations with HPC volume, we addressed the possibility of false negatives in our original approach by iterating the correlational analyses above after introducing three amendments: i) we fragmented the anterograde retrieval composite score into 4 composite scores (visual / verbal recall/recognition), taking into account the possibility of different relationships of recall vs. recognition memory scores with brain abnormalities; ii) we applied a more lenient correction for the number of structural/functional abnormalities (n=13), separately for each composite score examined (n=6; visual / verbal recall / recognition, remote autobiographical memory, visual forgetting; Supplementary Table 4); iii) in a post-hoc fashion, we examined the relationship of these memory scores with the manually delineated anterior vs. posterior HPC portions at uncorrected levels (Supplementary Table 5).’

Reviewer #2:

[…] These findings provide important linkages with other work in the literature connecting hippocampal compromise to structural and functional integrity of the thalamus and the posterior parietal cortex.In general, I found the paper well written, and that the methods and statistical analyses were conducted rigorously. There are a few methodological details that should be stated more clearly in the manuscript, which I have detailed below. I have some other comments and questions regarding the pattern of anatomical changes observed.

We would like to thank the reviewer for the encouraging feedback.

1) The finding that thalamic grey matter is reduced in these amnesic individuals is consistent with the literature on developmental amnesia (e.g. Dzieciol et al., 2017). Were the mammillary bodies examined in this dataset? It would be interesting to know if individuals who sustain hippocampal compromise during adulthood also develop atrophy within these structures. Alternatively, it would also be interesting to know if thalamic atrophy might occur in the absence of changes to the mammillary bodies, given that in the macaque and human there appear to be direct fornical projections from the hippocampus/subiculum to the anterior thalamus (Bubb et al., 2017).

We have not yet examined the integrity of the mammillary bodies in our structural MRI datasets. We agree that this is a very interesting question that is certainly worthy of attention. However, on reflection we feel that it might be better addressed in a follow-up manuscript, perhaps with additional reference to (i) forniceal integrity, (ii) also including manual delineation of anterior and posterior thalamic regions, (iii) since the emphasis of these analyses would be on mechanisms of damage rather than specific effects on behaviour, and (iv) since we have already taken into account several structural and functional abnormalities in examining their relationships with different aspects of memory impairment, and adding more is likely to increase the likelihood of false negatives. The following text has now been added to the Discussion section:

‘Finally, our finding that thalamic volume was reduced in our patient group is consistent with the literature on developmental amnesia due to early hypoxic-ischaemic encephalopathy, where HPC damage has been noted along with atrophy in the thalamus and the mammillary bodies [e.g. (Dzieciol et al., 2017)]. A question for future research is therefore whether adult-onset HPC damage is accompanied by atrophy in the mammillary bodies, or whether thalamic atrophy may occur in the absence of changes to the mammillary bodies, given the evidence for direct forniceal projections from the HPC/subiculum to the anterior thalamus in human and non-human primates (Bubb et al., 2017a).’

2) It is stated that inflammatory pathology would be 'expected to spread to adjacent areas, rather than to remote regions in the extended HPC network.' Is there empirical evidence for this assertion? Or, conversely, might there be evidence that inflammatory pathology could spread from the hippocampus to the thalamus via the fornix?

We agree with the reviewer that this assertion would require empirical evidence. As we are not aware of any findings in the literature to directly support this, we have now removed this assertion.

3) A bit more information about the different types of functional connectivity is needed. From my read of the paper, two types of functional connectivity were performed: resting state functional connectivity (rsFC) and resting-state amplitude of low frequency fluctuations (rsALFF). Why were these two methods used and what can one method offer that the other method cannot? This should be made clearer either in the methods or earlier in the manuscript.

In the Materials and methods section, we have now provided more information regarding the different types of analyses conducted on our resting-state fMRI datasets, along with further justification why rsALFF was also examined along with rsFC, as follows.

‘We explored resting-state functional abnormalities in patients with respect to both segregation and integration, i.e. in terms of hemodynamic activity in local regions and functional connectivity between regions. While the majority of rsfMRI studies investigate the correlations between brain areas from the perspective of integration, i.e. ‘resting-state functional connectivity’ (rsFC) (providing holistic information on sets of interacting brain regions) and the abnormal integration between brain areas in patient groups relative to healthy controls, these approaches do not directly provide information on the specific brain regions that show abnormalities in patients (in the form of the amplitude of spontaneous brain activity). The latter are reliably indexed by the resting-state amplitude of low-frequency fluctuation (rsALFF) of the rsfMRI signal, and help us identify the specific brain regions of abnormal spontaneous activity (Zang et al., 2007; Zou et al., 2008), similar to resting-state CBF and glucose metabolic rate in PET studies. For the above reasons, the combined application of these two methods (rsFC and rsALFF) has been held to provide more information than either method alone (Lv et al., 2018). We used data-driven approaches in both.’

4) Were mediation analyses performed for each of the correlations and only reported for the models in which either the thalamic or connectivity patterns significantly mediate the relationship between hippocampal volume and memory? In some parts of the paper these mediation analyses seem hypothesis driven (e.g. subsection “Anterograde memory: Visual Recognition”) whereas in other parts of the paper I wasn't sure if these mediation analyses were hypothesized a priori (e.g. subsection “Remote Autobiographical Memory”). This issue should be made clarified.

In the original version of our manuscript, mediation analyses were planned for the structure/function-behaviour relationships for each of the 6 memory composite scores (visual / verbal recall / recognition, visual forgetting, and remote autobiographical memory). Nevertheless, in the case of verbal recognition memory, inter-HPC rsFC did not satisfy the assumptions for conducting a mediation analysis – namely, HPC volumes did not correlate across patients with their reduced inter-HPC rsFC. Instead, we conducted a multiple step-wise linear regression analysis, to investigate whether HPC volumes explain a significant portion of the variance in addition to inter-HPC rsFC (multiple step-wise linear regression relies on the assumption that the independent variables are not correlated).

Regarding the mediation analysis reported in subsection “Remote Autobiographical Memory” of the original version of our manuscript, we wanted to investigate whether the observed abnormalities in the PCC are triggered by HPC damage by means of secondary thalamic atrophy – consistent with the idea of the HPC-diencephalic-cingulate network (Bubb et al., 2017b; Vann et al., 2009). However, since these analyses are only supplementary and do not directly address our main hypothesis (that certain aspects of memory impairment are mediated by network-wide abnormalities in the extended HPC system following HPC damage), we have moved these analyses and their discussion to Figure 2—figure supplement 1, in order to further enhance readability of our Results section (as also recommended by reviewer #1).

We have amended the beginning of our Results section with a flowchart (Figure 1) illustrating the different analyses undertaken, in order to enhance readability. We also provide a brief outline of the structure of our Results section, as follows:

‘Our analysis approach is summarized in Figure 1. We first (1) identified cognitive deficits by comparing patients with healthy controls in a broad range of tests of neuropsychological assessment. We then (2) identified regions in which patients showed structural and (3) functional abnormalities relative to healthy controls. We then (4) examined the relationship between structural / functional brain abnormalities and memory impairment across our patients.’

In our revised Introduction, we now highlight that mediation analyses were conducted with HPC volumes as the independent variable:

‘We predicted that:: i) HPC damage (documented to occur focally in autoimmune LE) would be accompanied by remote abnormalities in hubs of the extended HPC system, including the HPC-diencephalic-cingulate network (Aggleton, 2014; Bubb et al., 2017a); ii) variation in the extent of these network-wide abnormalities across patients would explain variability in memory impairment over and above HPC atrophy; iii) Moreover, on our assumption that HPC damage leads to broader network disruption, we predicted that relationships between HPC atrophy and memory impairment would be mediated by this broader network disruption. This prediction is in line with recent functional imaging and lesion studies that emphasize the importance of broader networks in episodic memory (Cook et al., 2015; Henson et al., 2016; Inhoff and Ranganath, 2017; Schedlbauer et al., 2015; Watrous et al., 2013); iv) taking these abnormalities into account would enable us to identify which particular aspects of memory impairment are a direct function of HPC atrophy.’

Reviewer #3:

[…] The main findings are that, while the patients have relatively focal volume loss within the hippocampus, the relationships between hippocampal volume and memory performance are usually mediated by relationships with other brain structures: notably functional connectivity with posterior midline regions in the case of anterograde memory and grey matter volume in the thalamus in the case of retrograde memory. I think these findings will make a useful contribution to the field of memory research, but I have a few issues that I feel the authors should address.

We would like to thank the reviewer for their encouraging feedback.

In the volumetric analyses, the authors seem to be testing whether the patient group as a whole has volume loss greater than 1.5 SDs below the mean of the controls. If correct, this seems to be extremely conservative and rather unusual. I would have expected a more direct comparison between the volumes of the patients and controls – with indications of the effect size of the difference. This is the approach done with the neuropsychological tests. It might help if the degree of volume loss is expressed as a percentage of the controls, with statistics relating to the direct comparison of the patients and controls. Perhaps I am missing something here?

We agree with the reviewer’s comment. reviewer 1 has made the same point. Our original approach had been adopted in keeping with standard practice in single-case and case-series studies, whereby HPC volumes of individual patients are shown to exceed the traditionally employed cut-off point of -1.5 SD from the mean of age-matched healthy controls, in the face of other MTL structures not exceeding this cut-off point. We have now adopted the quantification of HPC (and other MTL and subcortical) volumes according to the method the reviewer recommended, and we have iterated the analyses of structure/function-behavior relationships, after regressing volumes against age, sex, scan source, and TIV across participants (as already done for the mean signal of each VBM cluster in the original manuscript). We have also included the mean and SD of the% volume reduction for our patient group in Table 2. We clarify this in our revised Materials and methods section:

‘We compared healthy controls and patients on the volumes of all structures in a series of ANCOVAs, using age, sex, scan source [MAP (Memory and Amnesia Project), or OPTIMA (Oxford Project To Investigate Memory and Aging)], and total intra-cranial volume (TIV; derived from the unified segmentation procedure in SPM12; see below) as between-subjects covariates. Volumes that were reduced in our patient group were residualized against these four variables (z-res) and entered in a series of bivariate correlations with patients’ memory scores.’

The authors correlate MRI measures with composite measures of performance based on averaging across tests of anterograde memory in which patients showed impairment. This seems an inadequate justification for choosing which tests contribute to the composite measures. It is interesting that the patients are unimpaired on tests of face recognition memory, and as the authors point out, this is in keeping with other findings (e.g. Taylor et al., 2007; Bird and Burgess, 2008; Smith et al., 2014; Bird, 2017). This would constitute a reasonable justification for not including face recognition memory in a composite visual recognition memory measure in patients with hippocampal damage. However, the selection of measures based on performance alone is biased.

Our aim was to identify explanations of memory impairment in patients, without taking the extra hypothetical step of attempting to identify the neural correlates of normal performance on the tasks. In the same manner, we focused on the brain regions where analyses showed a group difference from controls. We thus correlated abnormal brain structure / function with abnormal behaviour. Since face recognition memory was unimpaired in patients, we did not include it in the composite score of visual recognition memory. We agree with the reviewer that, since normal face recognition is in keeping with findings from other groups, omitting it from the composite score is justified.

In the description of the methods for "Resting-state Hemodynamic activity: rsALFF" the authors define RsALFF as the total power within the frequency rage (0.01-0.1 Hz). However, in the pre-processing section they say that they apply a temporal band pass filter of 0.01-0.1 Hz. To me, this would appear to affect the signal that they want to analyse. Could the authors justify their procedure and confirm if this procedure is used by other groups reporting the results of RsALFF analyses?

We confirm that the majority of the papers reporting analyses of resting-state ALFF have employed the same preprocessing step, i.e. band-pass filtering to retain frequencies between 0.01Hz and 0.1 Hz. This procedure removes low-frequency drift and high-frequency respiratory and cardiac noise, before examining ALFF within the low-frequency range of oscillations (0.01-0.1 Hz) (e.g. (Cui et al., 2014; Dai et al., 2012; Satterthwaite et al., 2012; Yao et al., 2012; Yi et al., 2012)). We have now added a clarification in our Materials and methods section:

‘The same pre-processing pipeline was used for both rsALFF and rsFC analyses, including denoising with a temporal band pass filter (0.01-0.1 Hz), as in the majority of studies examining rsALFF [e.g. (Cui et al., 2014; Dai et al., 2012; Satterthwaite et al., 2012; Yao et al., 2012; Yi et al., 2012)].’

The study and its results are situated with respect to neural dysfunction at sites separated from the hippocampus. While this is very likely to be the cause of volume loss in the thalamus, it is not necessarily the case for the resting state fMRI results. It is possible that the patients engage different cognitive processes during the resting state scan compared to the controls – and therefore the differences in the engagement of the anterior and posterior midline structures reflects this. These regions are strongly implicated in "mind-wandering" – following undirected trains of thought. There is also evidence that patients with hippocampal damage differ markedly in the form and content of their mind-wandering compared with healthy adults (McCormick et al., 2018). It is also possible that the patients simply forgot what they were doing in the scanner, and spent more time attending to, e.g., the noise of the scanner than the controls did. My point is that functional differences between the patients and controls could reflect the BOLD engagement in different cognitive processes rather than more fundamental differences in neuro/vascular functioning in these regions per se. I would encourage the authors to discuss this issue.

The reviewer is making an important and interesting point, and we have now acknowledged this interpretation in our Discussion section:

‘It is important to acknowledge an alternative interpretation of the relationship that we observed between resting-state functional abnormalities and memory measures. McCormick and colleagues recently showed that HPC amnesics differ in the form and content of their ‘mind-wandering’ compared with healthy adults (McCormick et al., 2018). It is possible that ‘resting-state’ functional abnormalities in our patients actually reflect (and are perhaps therefore mediated by) differences between healthy controls and patients with respect to the extent of mind wandering in the scanner, rather than differences in neurovascular functioning in these regions per se. Anterior and posterior midline structures are strongly implicated in mind-wandering. Nevertheless, rsfMRI measures (predominantly rsFC) have repeatedly provided reliable correlates of memory impairment in HPC amnesia [e.g. (Heine et al., 2018; Henson et al., 2016)]. Moreover, this interpretation is not inconsistent with the basic premise of our argument, namely, that the effects of HPC damage on memory are mediated by other processes that are compromised following HPC damage. Our study was not designed to disambiguate the level of disruption that is mediating the effects of HPC atrophy. In other words, the disruption could be at the cognitive level, or at the neurovascular level, or at both levels. Further work is needed to disambiguate those two interpretations.’